# Linking spinal circuit reorganization to recovery after thoracic spinal cord injury

Natalia A Shevtsova[1], Andrew B Lockhart[1], Ilya A Rybak[1], David SK Magnuson[2,3], Simon M Danner[1]*

[1]Department of Neurobiology and Anatomy, Drexel University College of Medicine, Philadelphia, United States; [2]Department of Neurological Surgery, University of Louisville School of Medicine, Health Sciences Campus, Louisville, United States; [3]Kentucky Spinal Cord Injury Research Center, University of Louisville School of Medicine, Health Sciences Campus, Louisville, United States

## eLife Assessment

This **important** study describes a computational model of the rat spinal locomotor circuits and how they could be plastically reconfigured after lateral hemisection or contusion injuries to replicate gaits observed experimentally *in vivo*. Overall, the simulation results **convincingly** mirror the gait parameters observed experimentally. The model suggests the emergence of detour circuits after lateral hemisection, whereas after a midline contusion, the model suggests plasticity of left-right and sensory inputs below the injury.

*For correspondence:
smd395@drexel.edu

Competing interest: The authors declare that no competing interests exist.

## Abstract
Rats exhibit significant recovery of locomotor function following incomplete spinal cord injuries, albeit with altered gait expression and reduced speed and stepping frequency. These changes likely result from and give insight into the reorganization within spared and injured spinal circuitry. Previously, we developed computational models of the mouse spinal locomotor circuitry controlling speed-dependent gait expression (Danner et al., 2017; Zhang et al., 2022). Here, we adapted these models to the rat and used the adapted model to explore potential circuit-level changes underlying altered gait expression observed after recovery from two different thoracic spinal cord injuries (lateral hemisection and contusion) that have roughly comparable levels of locomotor recovery (Danner et al., 2023). The model reproduced experimentally observed gait expression before injury and after recovery from lateral hemisection and contusion and suggests two distinct, injury-specific routes to restored function. First, recovery after lateral hemisection required substantial functional restoration of damaged descending drive and long propriospinal connections, suggesting compensatory plasticity through formation of detour pathways. Second, recovery after a moderate midline contusion predominantly relied on reorganization of spared sublesional networks and altered control of supralesional cervical circuits, compensating for weakened propriospinal and descending pathways. Despite these differences, sensitivity analysis revealed that restored activation of sublesional lumbar rhythm-generating circuits and appropriately balanced lumbar commissural connectivity are the key determinants of post-injury gait expression, suggesting that injury symmetry shapes how the cord reorganizes, but effective recovery in both cases depends on re-engaging these lumbar networks, which makes them prime targets for therapeutic intervention.

## Introduction

Although incomplete spinal cord injuries partially disrupt communication across the lesion, they are often followed by substantial recovery of locomotor function (**Burns and Ditunno, 2001**; **Kuerzi et al.,**

**Figure 1.** Model concept. (**A**) Organization of the spinal locomotor circuitry (intact). (**B**, **C**) Neural structures affected by thoracic hemisection (**B**) or contusion (**C**) injuries. Spheres represent neural populations involved in commissural and long propriospinal pathways. Descending drives and synaptic interactions are shown by arrowheads. Decreased color intensity in (**C**) signifies partial disruption of pathways by contusion injury. CINs, commissural interneurons; LPNs, long propriospinal interneurons.

*2010*). These improvements are accompanied by reorganization of spared and severed descending pathways and spinal circuitry (*Raineteau and Schwab, 2001*; *Fouad and Tse, 2008*; *Takeoka et al., 2014*; *Filli and Schwab, 2015*; *Engmann et al., 2020*; *Lemieux et al., 2024*). However, it remains unclear which plastic changes of neural connectivity underlie functional recovery and if the symmetry of the lesion influences those changes.

Locomotion is primarily controlled by spinal circuitry (*Grillner, 1981*; *Orlovsky et al., 1999*; *Kiehn, 2006*; *Kiehn, 2016*; *McCrea and Rybak, 2008*; *Rybak et al., 2015*), which produces the locomotor rhythm and pattern, and controls interlimb coordination in response to supraspinal inputs. Flexor–extensor alternation in each limb is regulated by rhythm generators (RGs) located bilaterally within the lumbar and cervical enlargements (*Kato, 1990*; *Ballion et al., 2001*; *Juvin et al., 2012*; *Rybak et al., 2015*; *Danner et al., 2016*; *Danner et al., 2017*; *Frigon, 2017*; *Figure 1A*). These RGs receive descending inputs from the brainstem, which modulate locomotor frequency and speed (*Danner et al., 2016*; *Capelli et al., 2017*; *Caggiano et al., 2018*; *Josset et al., 2018*; *Ausborn et al., 2019*), and are interconnected by commissural (left–right) and long propriospinal (lumbar–cervical) neuronal pathways, which define interlimb coordination (*Talpalar et al., 2013*; *Danner et al., 2016*; *Danner et al., 2017*; *Ruder et al., 2016*). Thoracic spinal cord injuries disrupt these descending pathways and inter-enlargement long propriospinal neuronal (LPN) connections, while leaving the sublesional lumbar and supralesional cervical circuitry largely intact (*Figure 1B and C*). Thus, the animals' ability for speed-dependent gait expression following post-injury recovery can provide insights into the state of reorganization of descending and inter-enlargement pathways (*Shepard et al., 2021*; *Shepard et al., 2023*; *Danner et al., 2023*).

Indeed, following thoracic lateral hemisection or moderate midline contusion injuries, rats recover weight-bearing overground locomotion, though with a reduced maximal speed and altered speed-dependent gait patterns (*Danner et al., 2023*). Specifically, post-hemisection recovery allows expression of a subset of pre-injury gaits – walk, trot, canter, and gallop – in a speed-dependent manner, albeit with reliance on the contralesional limb as lead in non-alternating gaits. In contrast, mild-to-moderate

midline contusion injuries result in a more pronounced speed reduction, loss of non-alternating gaits, and emergence of novel alternating gait patterns (*Danner et al., 2023*).

We developed a series of computational models of the spinal circuitry that controls speed-dependent interlimb coordination and gait expression (*Danner et al., 2016*; *Danner et al., 2017*; *Ausborn et al., 2019*; *Ausborn et al., 2021*; *Zhang et al., 2022*). By reproducing the speed-dependent gait patterns of intact mice (*Bellardita and Kiehn, 2015*; *Lemieux et al., 2016*) and simulating the effects of silencing or ablating various classes of commissural (*Talpalar et al., 2013*; *Bellardita and Kiehn, 2015*) and long propriospinal interneurons (*Ruder et al., 2016*; *Zhang et al., 2022*), these models offer insights into the organization and function of spinal locomotor circuitry, including the roles of commissural and long propriospinal interneurons, as well as the broader interactions between brainstem drive and the spinal circuitry that regulate gait expression. Given the similarity in gait patterns between rats (*Danner et al., 2023*) and mice (*Bellardita and Kiehn, 2015*; *Lemieux et al., 2016*), we adapted these models to rats as a foundation for studying post-injury plasticity.

Here, we use a computational model of spinal circuitry to investigate potential reorganization of neural connectivity underlying locomotor control following recovery from symmetrical thoracic contusion and asymmetrical (lateral) hemisection injuries. The model reproduces speed-dependent gait expression in rats before injury and after recovery from hemisection and contusion. In our model, reproduction of altered gait expression observed after recovery from the fully asymmetrical hemisection injury required substantial functional restoration of severed long propriospinal connections and descending drive. This suggests the involvement of compensatory mechanisms, such as local axon sprouting and rewiring, resulting in formation of detour pathways via the intact hemicord. In contrast, after a symmetrical contusion injury, the model suggests that the observed post-recovery gait changes rely on strengthened intra-enlargement connectivity, increased afferent input, and/or changes in intrinsic excitability of the sublesional circuitry that compensate for weakened long propriospinal connections and descending drive.

## Results
### Model of the rat spinal locomotor circuitry
The intact (pre-injury) model (*Figure 1* and *Figure 2*) was based on our previous models of mouse speed-dependent gait expression (*Danner et al., 2017*; *Zhang et al., 2022*) and was adapted to reproduce rat locomotion. Like the previous models, it consists of four RGs, located on the left and right sides of the cervical and lumbar compartments (left and right fore and left and right hind RGs). Each RG includes a flexor and extensor center (F and E) that mutually inhibit each other via inhibitory interneuron populations (InF and InE). Both flexor and extensor centers incorporate a persistent (slowly-inactivating) sodium current ($I_{NaP}$, see Methods), allowing them to intrinsically generate rhythmic activity under certain conditions defined by the level of excitation, which is regulated by descending brainstem drive. The flexor centers operate in a bursting mode, while the extensor centers receive relatively high drive that keeps them in the mode of tonic activity and exhibit rhythmic activity only due to rhythmic inhibition from the corresponding flexor centers. Thus, each RG produced alternating flexor–extensor activity defining the two major locomotor phases (flexor and extensor). The frequency of RG oscillations was defined by brainstem drive to each flexor center; it increased with increasing drive, primarily due to shortening of the extensor phase durations. This reflects the asymmetric modulation of stance and swing phase durations observed in limbed locomotion across species (*Halbertsma, 1983*; *Hildebrand, 1989*; *Bertram, 2016*), including rats (*Cohen and Gans, 1975*; *Danner et al., 2023*), where increases in locomotor frequency are primarily achieved by shortening stance (extensor) phases while swing (flexor) phases remain relatively stable – a pattern also generated intrinsically by spinal locomotor circuits across species (*Frigon and Gossard, 2009*; *Danner et al., 2015*; *Shevtsova et al., 2015*).

The left and right RGs within the cervical and lumbar compartments interact via a series of homologous commissural pathways mediated by commissural ($V0_D$, $V0_V$, and V3) and local (V2a, Ini, and InE1) interneurons, coordinating the activities of the left and right homologous RGs. The cervical and lumbar RGs interact via a series of homolateral (Sh2 and LPNi) and diagonal ($dV0_D$, $dV0_V$, and aV3) long propriospinal pathways, coordinating the fore–hind RG activities. Homologous (intra-enlargement) $V0_D$ and $V0_V$ and descending diagonal $V0_D$ interneurons receive inhibitory brainstem drive, allowing

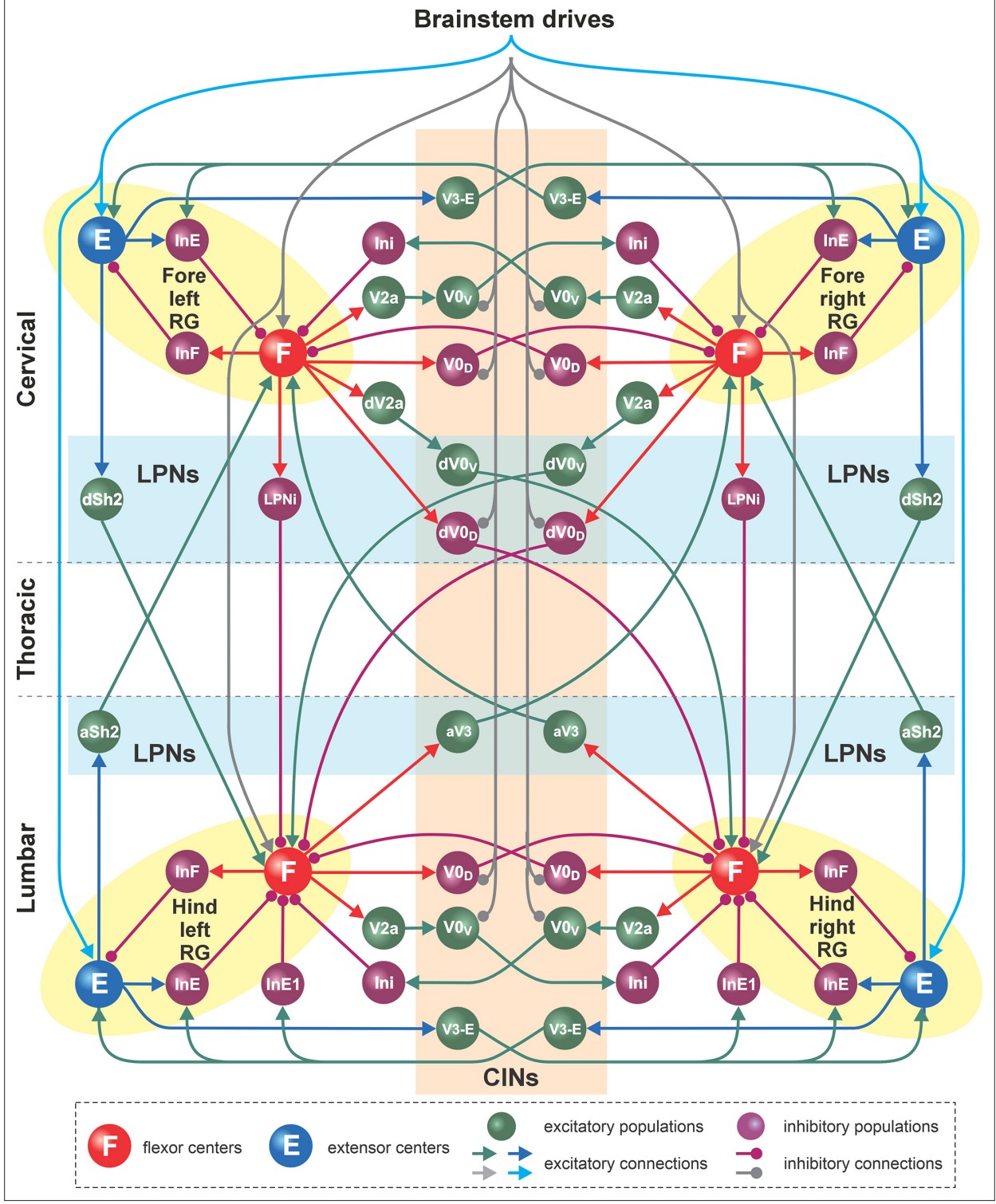

**Figure 2.** Detailed model schematic. Spheres represent neural populations. Excitatory drives and connections are marked by arrowheads; inhibitory connections are marked by circles. RG, rhythm generator; In, interneuron; CINs, commissural interneurons; LPNs, long propriospinal neurons; a-, ascending; d-, descending. See text for details. Adapted from *Danner et al., 2017*, and *Zhang et al., 2022*, with modifications.

them to coordinate left–right and fore–hind activities in a speed-dependent manner. The organization of spinal locomotor circuitry was based on multiple sets of experimental findings on speed-dependent gait expression of intact mice and after manipulating various classes of local, commissural, and long propriospinal interneurons (*Talpalar et al., 2013*; *Bellardita and Kiehn, 2015*; *Rybak et al., 2015*; *Shevtsova et al., 2015*; *Danner et al., 2016*; *Danner et al., 2017*; *Danner et al., 2019*; *Ruder et al., 2016*; *Ausborn et al., 2021*; *Zhang et al., 2022*). The detailed description of the proposed connectome of spinal circuitry can be found in *Danner et al., 2017*, and *Zhang et al., 2022*. The connection weights and neuron parameters in the current model were adjusted to reproduce characteristics of locomotor behavior in rats (see Methods).

We quantified frequency-dependent changes in inter-RG coordination and flexor/extensor phase durations in response to variations in brainstem drive (parameter $\alpha$, see Methods) by calculating normalized phase differences between activities of pairs of RGs and compared the resulting gait expression to rat locomotion. Specifically, inter-RG coordination in the model was compared to inter-limb coordination in rats, flexor and extensor phase durations were compared to swing and stance phase durations, and the period of oscillation was compared to the step-cycle duration in rats. These parameters were used to calculate gait in both the model and in rats to assess frequency-dependent gait expression (see Methods).

## Model reproduces gait expression of intact rats

To ensure suitability of the model to study potential circuit reorganization following recovery from spinal cord injury, we first compared speed-dependent gait expression of the model in the intact case (intact or pre-injury model) with that of intact rats while traversing a 3 m long tank (*Danner et al., 2023*). Locomotor bouts consisting of periods of acceleration and deceleration were simulated by successively and repetitively ramping the brainstem drive (parameter $\alpha$, see Methods) from a value at which trot is usually expressed ($\alpha$=0.55) to one at which bound is expressed ($\alpha$=1.05) and back down; step-to-step variability was ensured by including a noisy current ($\sigma_{\text{Noise}}$ = 1.1 pA; see Methods).

Representative examples of bouts of locomotor activities of the intact model and rat are shown in *Figure 3A1 and A2*, respectively. Step frequencies increased progressively in both the model and rats, driven primarily by shortening of the extensor and stance phases (*Figure 3A1 and A2*, upper panels). As frequency increased, the hind left–right normalized phase differences in the model and rats (orange lines in *Figure 3A1 and A2*, bottom panels) gradually shifted from alternation (~0.5 phase difference) to synchronization (~0/1 phase difference). Meanwhile, homolateral phase differences remained close to alternation (green), and diagonal phase differences transitioned from synchronization to alternation (blue). As a result, both the model and rat transitioned from trot to gallop and then half-bound gallop with increasing frequency. As shown previously (*Danner et al., 2023*), speed-dependent gait changes were gradual, rather than abrupt, resulting in the expression of transitional gaits (*Figure 3A*).

The average gaits identified in the intact model (*Figure 3B1*) – trot, canter, transverse and half-bound gallops (with left and right lead), and bound — closely matched those observed in rats (*Figure 3B2*). In the model, trot was characterized by synchronized diagonal RG activity and alternating left–right activation of homologous fore and hind RGs. Canter exhibited synchronization of only one pair of diagonal RGs. Gallops showed quasi-synchronization of the hind RGs and a nonzero phase difference between the fore RGs. Bound was characterized by left–right synchronization of the fore and hind RGs and alternation of the homolateral and diagonal RGs. Note the striking similarity of average extensor/stance phases and normalized phase differences in modeling and experimental results (*Figure 3B1 and B2*).

To compare frequency-dependent gait expression between the model and rats across bouts, normalized phase differences between pairs of RGs (*Figure 4A1*) and corresponding pairs of limbs (*Figure 4A2*) were plotted against locomotor frequency. In both cases, the frequency-dependent gait transitions from trot through gallop to half-bound gallop and bound (*Figure 4B1 and B2*) can be seen as progressive changes in the normalized phase differences (*Figure 4A1 and A2*). At low frequencies (~3–4 Hz), left–right alternation of the hind RGs/hindlimbs and the fore RGs/forelimbs dominated. At medium frequencies (~4–5.5 Hz), left–right phase differences showed progressive change from alternation to synchronization in both directions up to essential synchronization, while some steps still maintained left–right alternation. Finally, at the highest frequencies (>6.5 Hz in model and >6 Hz in rats), most steps exhibited left–right fore and hind phase differences close to 0 or 1 (synchronized

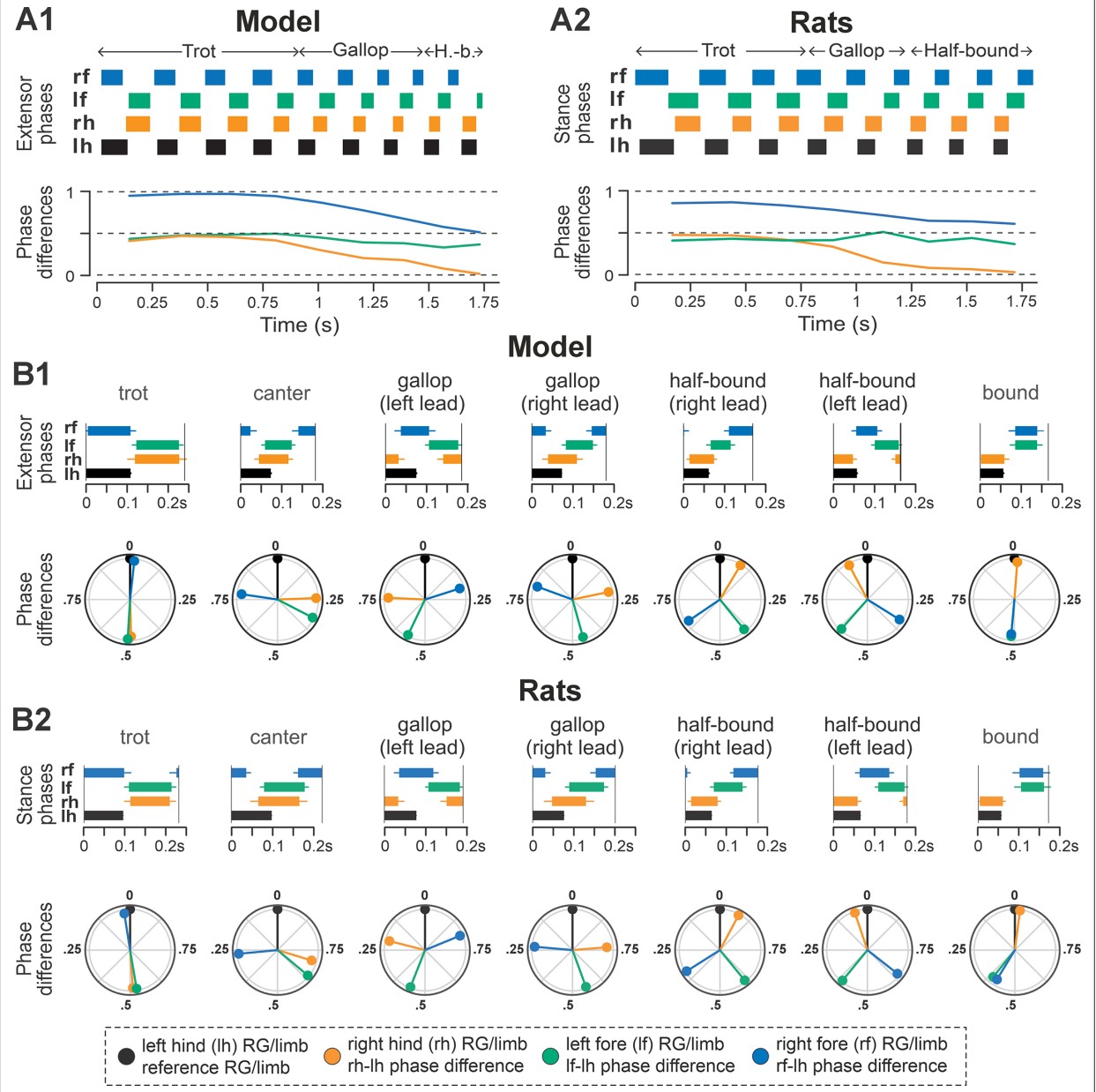

**Figure 3.** Gait expression in pre-injury/intact model and rats. (**A1**, **A2**) Extensor/stance phases (upper panels) and instantaneous normalized phase differences (bottom panels) of representative bouts for the model (**A1**) and a rat (**A2**). (**B1**, **B2**) Average extensor/stance phases for each gait (upper panels; error bars indicate circular standard deviations) and circular plots of average normalized phase differences for each gait (bottom panels; vector length corresponds to mean resultant length, R) expressed in the intact model (**B1**) and rats (**B2**). Detailed statistical results for rats are reported in *Danner et al., 2023*. H.-b., half-bound; RG, rhythm generator.

activity). The normalized homolateral phase differences remained alternating, close to 0.5 across the frequency range. The normalized diagonal phase differences gradually changed from synchronized to alternating. In parallel, phase durations showed asymmetric frequency-dependent shortening, with a more pronounced reduction in extensor (stance) phase duration relative to flexor (swing) phase duration, consistent between the model (*Figure 4C1*) and rats (*Figure 4C2*).

Trot in the intact model was expressed at oscillation frequencies from about 3 to 6 Hz (*Figure 4B1*) and was the most prevalent gait in our simulations (~45%, *Figure 5A1*), which is similar to the intact rats (*Figure 4B2* and *Figure 5A2*). The transverse gallop, both in the model and in rats, was expressed at slightly higher frequencies (~4.5–6.5 Hz) and accounted for ~30% of all steps, making it the second

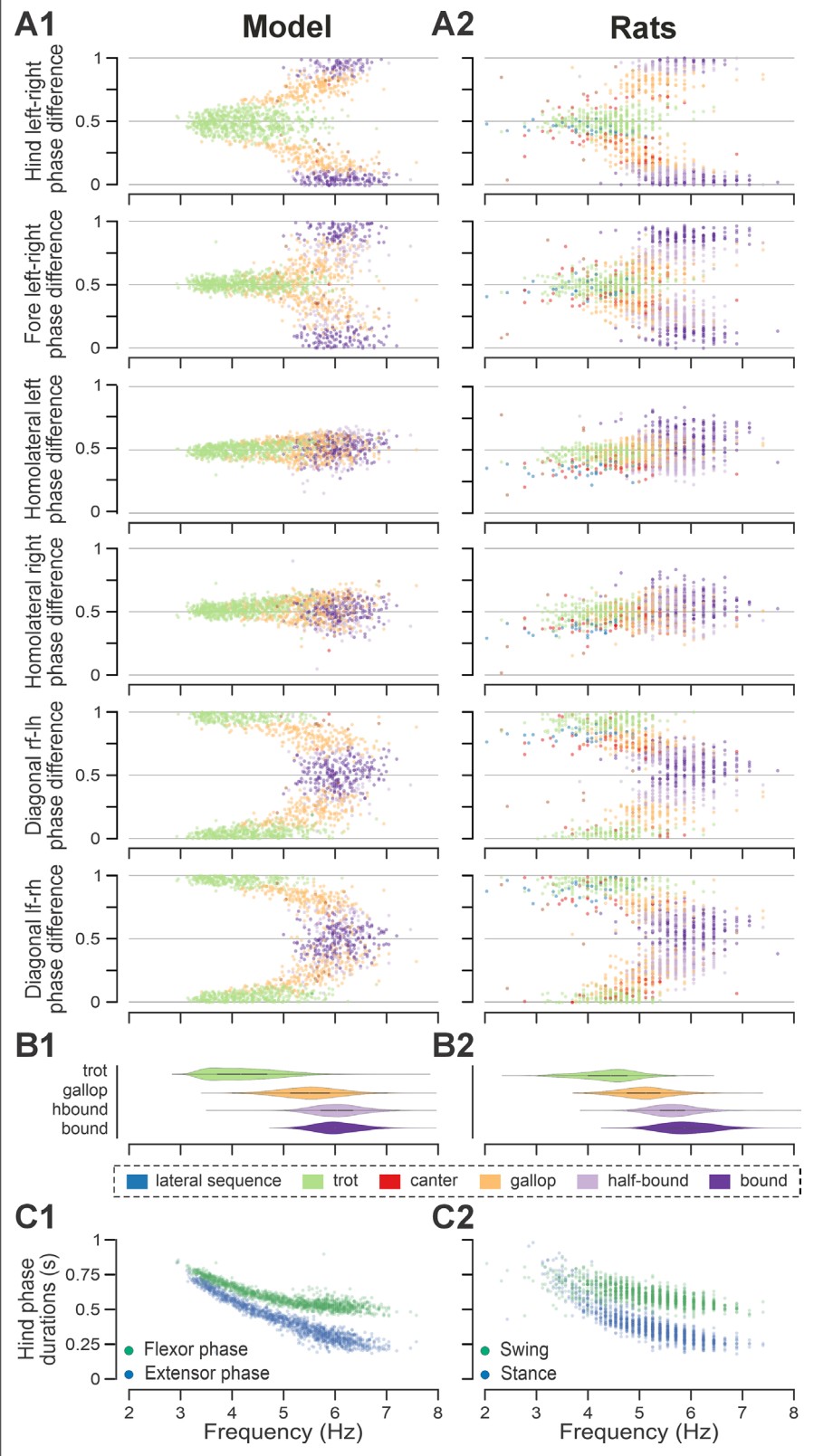

**Figure 4.** Frequency-dependent distribution of normalized phase differences, gaits, and phase durations in the model and in rats. (**A1**, **A2**) Scatter plots of normalized phase differences against frequency of locomotor oscillations in the intact model (**A1**) and rats (**A2**). Each dot represents one step cycle. Gaits are classified for each step cycle and color-coded. (**B1**, **B2**) Distribution of gaits vs. locomotor frequency in the model (**B1**) and

*Figure 4 continued on next page*

*Figure 4 continued*

rats (**B2**). Due to the low prevalence of lateral-sequence and canter steps, these gaits were omitted in (**B1**) and (**B2**). (**C1**, **C2**) Flexor and extensor phase duration in the model (**C1**) and duration of swing and stance in rats (**C2**) against frequency of locomotor oscillations. The same number of step cycles is shown for the model and animals; model step cycles were randomly sampled. l-, left; r-, right; -f, fore RG/limb; -h, hind RG/limb; RG, rhythm generator.

most prevalent gait (*Figure 4B1, B2* and *Figure 5A1, A2*). Half-bound gallop was observed in the model at ~5 to ~7 Hz (*Figure 4B1*), similar to rats (*Figure 4B2*), and was observed in ~10% of all steps (*Figure 5A1 and A2*). Bound was expressed at ~5–7 Hz in both the model and rats and accounted for ~10% of steps (*Figure 4B1, B2* and *Figure 5A1, A2*). Canter was the least prevalent gait in the intact model (~0.5% of steps; not shown in *Figure 4B1*), and also occurred rarely, mostly as a transitional gait, in rats (*Danner et al., 2023*). Although the exact proportions of the expressed gaits in the intact model and rats differ (*Figure 5A1 and A2*), their frequency-dependent distribution is qualitatively similar (compare *Figure 4B1 and B2*). Importantly, both in the model and intact rats, the ranges of locomotor frequency where particular gaits were expressed overlapped (*Figure 4B1 and B2*) and transitions between gaits occurred most prevalently between gaits in the sequence of their frequency-dependent expression (*Figure 4B1* and *Figure 5C1, D1*), similar to the gait transitions in the intact rats (*Figure 4B2* and Figure 2 in *Danner et al., 2023*).

Finally, to assess the step-to-step variability while accounting for variance introduced by gait changes, we measured the mean deviations from the circular exponential moving averages of each phase difference per bout (*Danner et al., 2023*). Both the model and rats exhibited a similar pattern: the mean deviations of the moving averages of the two left–right and diagonal phase differences were comparable to each other and consistently higher than those of the two homolateral phase differences (*Figure 5B1 and B2*).

In summary, the model reproduces the experimental results on locomotion of intact rats well: both the model and rats show progressive gait transitions with frequency increase (from trot to gallop and then to half-bound gallop and to bound) on a continuum in phase space, and gaits were expressed in overlapping frequency ranges.

## Modeling recovery from lateral thoracic hemisection injury

Following recovery from the asymmetrical lateral thoracic hemisection, rats regain the ability to locomote over a wide range of speeds but lose the ability to perform the fastest gaits (half-bound gallop and bound) and exhibit clear left–right asymmetry in interlimb coordination, predominantly using the contralesional limb as the lead during asymmetric gaits such as canter and gallop (*Danner et al., 2023*). Nevertheless, the expressed gaits represent a subset of intact gaits with appropriate interlimb coordination. Here, we used the model to investigate potential reorganization of spinal locomotor circuitry that could underlie the observed speed-dependent gait expression following recovery from hemisection.

To do this, we first modeled the acute effect of the injury by eliminating severed connections from the model (*Figures 1B and 6A*). Specifically, right thoracic hemisection was modeled by eliminating brainstem drive to the ipsilesional lumbar compartment and all LPN connections on the side of the injury. Note that axons of contralaterally projecting LPNs generally cross the midline at a spinal level close to the cell body (*Reed et al., 2006*; *Pocratsky et al., 2020*); thus, we assumed that contralaterally projecting LPNs with contralesional cell bodies and homolaterally projecting LPNs with ipsilesional cell bodies are axotomized by the thoracic hemisection.

Then, by manually tuning model parameters, we sought to identify a small set of connectivity changes sufficient to reproduce speed-dependent gait expression observed experimentally in rats following recovery from hemisection injury (*Figure 6B*). Specifically, we assumed that the injury-affected LPN and descending pathways, along with local cervical and lumbar circuitry, undergo plasticity associated with functional recovery, e.g., through regrowth (sprouting), detour pathways, synapse loss/formation, or upregulation of sensory feedback, and that descending control of lumbar and cervical circuitry through uninjured pathways is altered.

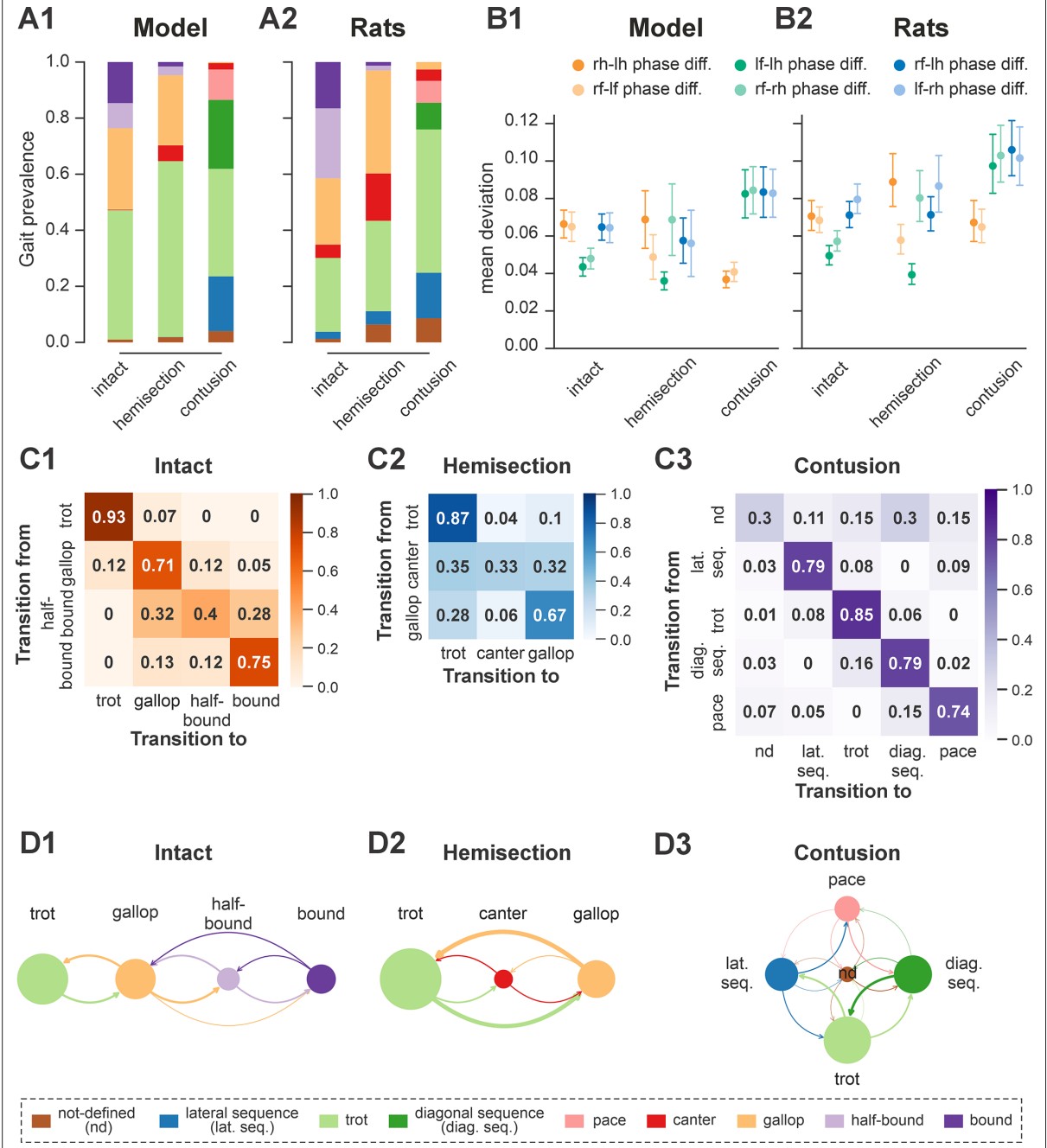

**Figure 5.** Gait prevalences, variability of interlimb coordination, and gait transition probabilities in model and rats. (**A1**) Prevalence of gaits in the intact model and following simulated recovery from hemisection and contusion injury. (**A2**) Prevalence of each gait across intact rats and rats after recovery from hemisection and contusion injury (recalculated from *Danner et al., 2023*). (**B1, B2**) Means of the deviations from the moving average of each phase difference for intact case and after recovery from hemisection and contusion injuries for the model (**B1**; error bars denote standard deviations) and rats (**B2**; error bars denote 95% confidence intervals). Detailed statistical results for rats are reported in *Danner et al., 2023*. (**C1–C3**) Matrices of gait transition probabilities in the intact model and following recovery from hemisection and contusion injury. (**D1–D3**) Gait transition graphs, where nodes represent gaits (size is proportional to their prevalence) and edges represent gait transitions (line widths are proportional to their frequency of occurrence). l-, left; r-, right; -f, fore RG/limb; -h, hind RG/limb; nd, not defined; diag., diagonal; lat., lateral; seq., sequence; RG, rhythm generator.

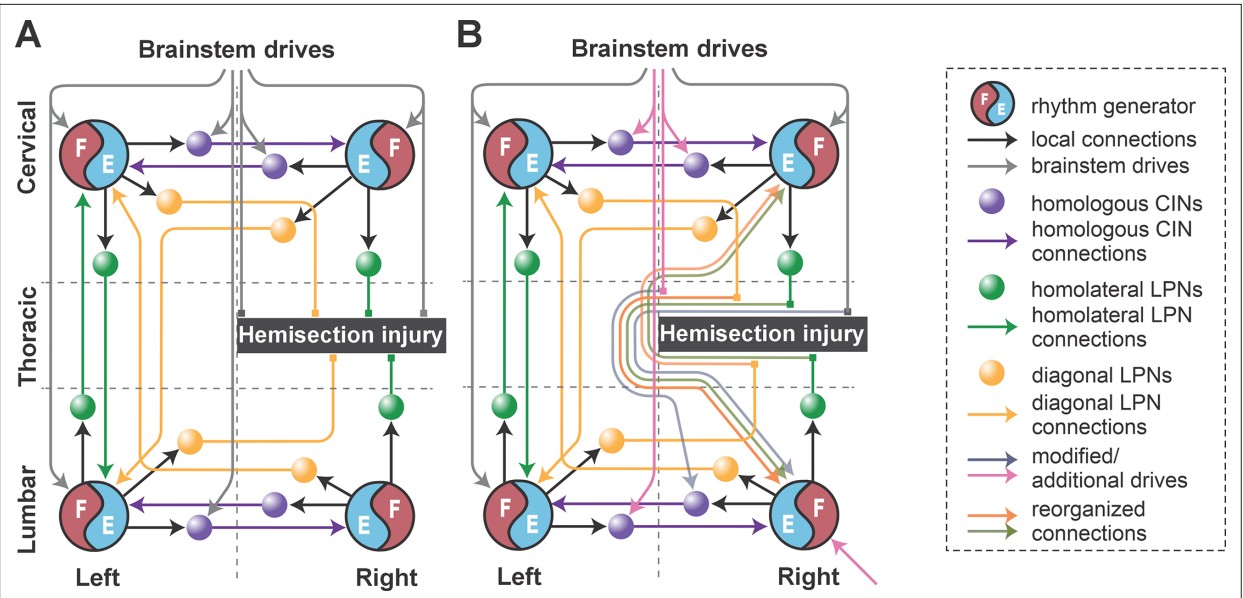

**Figure 6.** Conceptual schematic of the impact of hemisection injury on the spinal locomotor circuitry (**A**) and its reorganization after recovery from hemisection predicted by the model (**B**). In (**B**), affected long propriospinal neuron (LPN) connections recovered functionally through detour pathways. Drives to lumbar and cervical commissural interneurons (CINs) were altered to strengthen left–right alternation. Drive to the ipsilesional lumbar rhythm generator (pink arrow at the bottom) was substituted by regenerated brainstem input and/or afferent feedback.

These assumptions were implemented in the model (*Figure 6B*) by:

1. (functional) recovery of the injury-affected LPN connections to 40% of their pre-injury weights,

2. recovery of drive to the ipsilesional hind RG to 90% of its pre-injury strength (this might be partially due to concomitant upregulation of sensory input),

3. recovery of inhibitory drive to the ipsilesional lumbar $V0_V$ commissural interneuron (CIN) to 50% of its pre-injury strength, and

4. reduction of inhibitory drive to cervical and contralesional lumbar $V0_V$ CINs to the same strength (50%), which through disinhibition strengthened left–right alternation.

The post-hemisection model exhibited frequency-dependent expression of interlimb coordination and gait patterns qualitatively similar to those observed in rats following recovery from hemisection (*Figures 5, 7, and 8* and Figure 2 in *Danner et al., 2023*). Exemplary bouts of locomotor activities from both the model and animals showed progressive transitions from trot to canter, followed by transverse gallop and back to trot (*Figure 7A1 and A1*, upper panels). In both the model and in rats, hind left–right normalized phase differences during gallop and canter were above 0.5, signifying the use of the left-lead variants of both gaits (*Figure 7A1 and A2*, bottom panels). Across all steps, average extensor phase timing and the normalized phase differences for each gait were qualitatively similar between the modeling and experimental results (*Figure 7B1 and B2*), with the most noticeable difference being the left–right hindlimb phase difference during trot, which in the model showed a stronger asymmetry, i.e., was shifted farther from alternation.

In both the model and in rats, the maximal locomotor frequencies were slightly reduced (compared to the pre-injury case shown in *Figure 4*) to approximately 6–6.5 Hz (*Figure 8*), and the high-frequency, synchronized gaits (half-bound gallop and bound) were expressed in very few steps (see also *Figure 5A1, A2* and *Figure 8*). On the other hand, the occurrence of canter notably increased compared to the intact case (*Figure 5A1 and A2*) and was distributed across a wide range of generated frequencies (*Figure 8*). In both the model and rats, trot (62% of all steps in the model and ~32% in rats) and gallop (~25% in model and ~36% in rats; *Figure 5A1 and A2*) remained prevalent and were also more stable (high probability of subsequent steps being of the same gait) than canter (Figures 5C2, D2, 2H2, I2 in *Danner et al., 2023*). Canter often occurred as a transitional gait (high

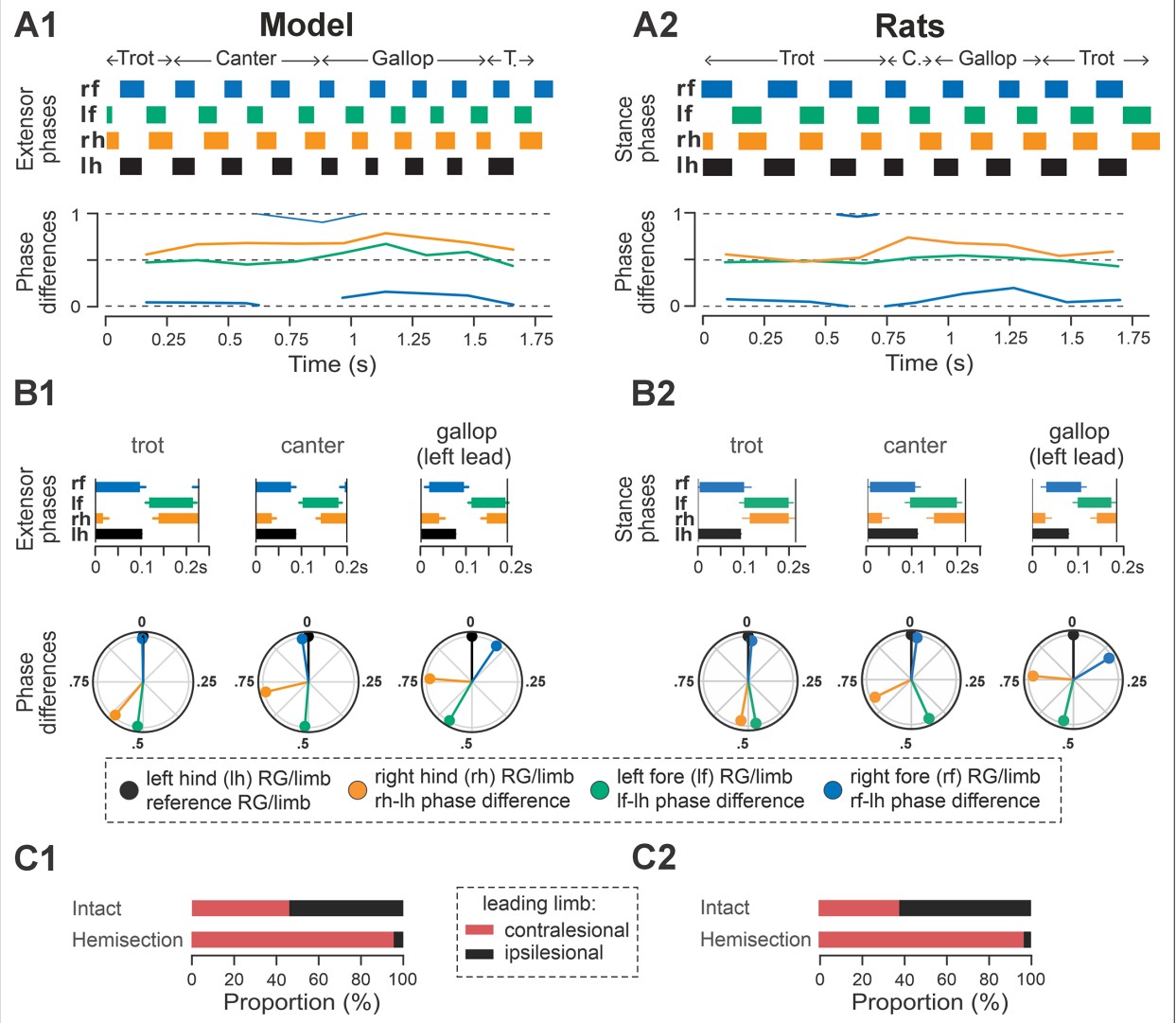

**Figure 7.** Gait expression in the model and rats following recovery after hemisection. (**A1**, **A2**) Extensor/stance phases (upper panels) and instantaneous normalized phase differences (bottom panels) of representative bouts for the model (**A1**) and a rat (**A2**). (**B1, B2**) Average extensor/stance phases (upper panels; error bars indicate circular standard deviations) and circular plots of average normalized phase differences for each gait (bottom panels; vector length corresponds to the mean resultant length, R) expressed in the post-hemisection model (**B1**) and rats (**B2**). Detailed statistical results for rats are reported in *Danner et al., 2023*. (**C1**) Prevalences of lead RG in the intact model and following simulated recovery after hemisection for gallop and canter. (**C2**) Prevalence of leading limbs (left or right forelimb that touches down second) pre-injury (intact) and after recovery of hemisection for gallop and canter in rats. Adapted from *Danner et al., 2023*. T., trot; C., canter; RG, rhythm generator.

probability that next step is of a different gait) and was frequently skipped at transitions from trot to gallop or gallop to trot (Figures 5C2, D2, 2H2, I2 in *Danner et al., 2023*).

The frequency-dependent distribution of phase differences across bouts (*Figure 8A1 and B1*) exhibits several features consistent with experimental observations in rats following recovery from hemisection injury (*Figure 8A2 and B2*). All steps were expressed on a continuum of phase differences that appears to be a subset of the phase differences of their pre-injury counterparts (*Figure 4A1 and B1*). Hind left–right normalized phase differences showed a progressive deviation from alternation (~0.5 phase difference) to quasi-synchronization (phase difference close to 0/1) that occurred only in the direction of the left-lead canter and gallop; right-lead canter and gallop were almost completely lost (*Figure 7C1 and C2*). Hind left–right synchronization (phase difference close to 0/1; characteristic for half-bound gallop and bound) was almost never reached. Furthermore, there was a left–right asymmetry of the homolateral phase differences: the ipsilesional homolateral normalized

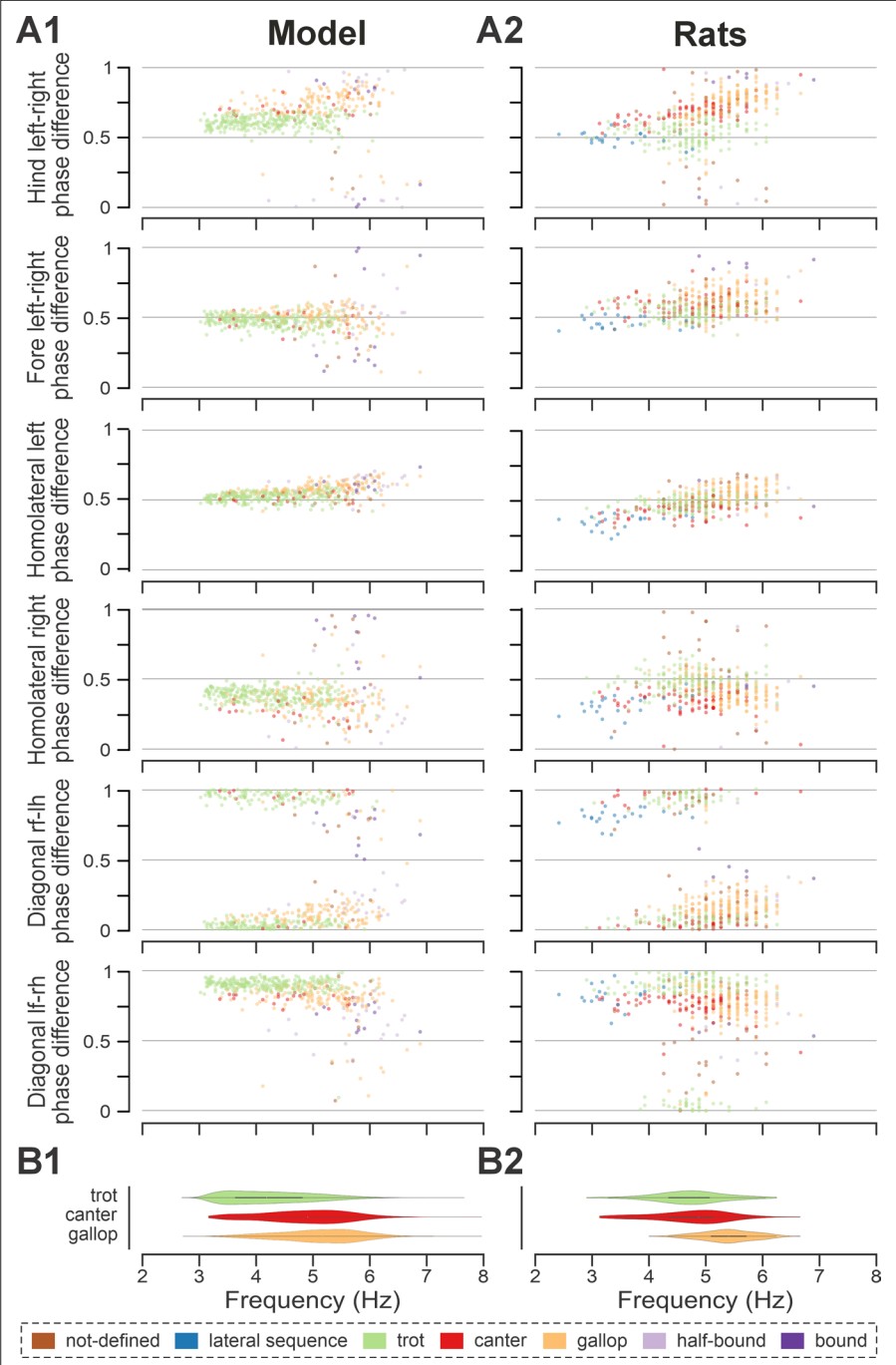

**Figure 8.** Frequency-dependent distribution of normalized phase differences in the model and in rats following recovery after hemisection. (**A1**, **A2**) Scatter plots of normalized phase differences are plotted against frequency of locomotor oscillations. Each dot represents one period/step cycle. Gaits are classified for each period/step cycle and color-coded. (**B1, B2**) Distribution of gaits vs. locomotor frequency in the post-hemisection model (**B1**) and rats (**B2**). The same number of step cycles is shown for the model and animals; model step cycles were randomly sampled. l-, left; r-, right; -f, fore RG/limb; -h, hind RG/limb; RG, rhythm generator.

phase differences exhibited lower values compared to pre-injury and to the contralesional homolateral phase differences.

Finally, the model exhibited changes in variability of phase differences relative to the pre-injury condition that were similar to those observed in rats (*Figure 5B1 and B2*). Specifically, compared to pre-injury, the variability (mean deviation from their moving average) of fore left–right and the

contralesional (left) homolateral phase differences decreased, whereas variability of the ipsilesional (right) homolateral phase difference increased. The variability of the other phase differences remained at a similar level to their pre-injury values, with a slight increase in the hind left–right phase difference and a slight decrease of the diagonal ones. Although the hind left–right phase difference increased more strongly in rats (*Danner et al., 2023*), the model results overall are in qualitative agreement with the experiments. It is interesting to note here that the variability of the contralesional (left) homolateral phase difference decreased in the model even though the contralesional LPN connection weights remained at pre-injury levels.

Altogether, these results show that the suggested reorganization of long propriospinal connectivity and descending drive allowed the model to reproduce key features of rat gait expression following recovery from thoracic hemisection injury: expression of a subset of pre-injury gaits, loss of the highest-speed gaits, and pronounced left–right bias evident in the preferential use of contralesional RG/limb as lead during left–right asymmetric gaits.

## Mechanism underlying left–right bias after hemisection

Recovered locomotion following hemisection injury was modeled as a partial (functional) recovery of the pathways severed by the injury and, thus, the model included asymmetries of LPN connectivity, as well as descending drive (*Figure 1B* and *Figure 6*). To investigate the contributions of these structural asymmetries on the emergence of the left–right bias and loss of high-speed gaits in the post-hemisection model, we created bifurcation diagrams of the normalized phase differences for the brainstem drive parameter $\alpha$ (*Figure 9*). These simulations were performed at a low level of noise ($\sigma_{\text{Noise}}$=5 fA, see Methods) to ensure identification of stable trajectories.

In the intact case, the model generated stable rhythmic locomotor-like activity when parameter $\alpha$ was changed in a stepwise manner from ~0.1 to 1.1 (*Figure 9A1*), which led to an increase in locomotor frequency from ~1 to ~7 Hz (*Figure 9B1*). Blue and red lines in *Figure 9A1* indicate the stable phase differences with increase or decrease in $\alpha$, respectively. Discrepancies between the red and blue lines (when ~$0.8<\alpha<0.92$) indicate regions of hysteresis and multi-stability. For these $\alpha$ values, multiple stable solutions coexisted and could be expressed depending on the initial conditions. The increased frequency was accompanied by sequential gait changes from lateral-sequence steps to trot to gallop and then half-bound gallop and bound (*Figure 9A1*). In the region of multi-stability, both trot and gallop were stable. Importantly, for the left–right asymmetric gaits (gallop and half-bound gallop), where the hind and fore left–right normalized phase differences are not at 0.5 or 0/1, both left–right symmetric solutions were stable. This corresponds to the stability of gaits with left and right lead RGs. The gait transitions and the corresponding bifurcations in the intact model are similar to our previous mouse models and are described in detail in *Danner et al., 2017*. The frequency range and step cycle compositions (relative phase durations) for all gaits, as well as gait expression, were consistent with rat locomotion (*Danner et al., 2023*).

In the case of simulated hemisection, the bifurcation diagrams show (*Figure 9A2*) that the range of brainstem drive parameter $\alpha$, in which the model is stable, was reduced compared to the intact case. This resulted in a reduction of the maximal frequency to ~6 Hz (*Figure 9B2*). With increasing drive, the model transitioned from a gait in-between lateral-sequence and canter to trot and then gallop (*Figure 9A2*). Half-bound gallop and gallop were lost as stable solutions, and all phase differences lost their second branches. The hind left–right phase difference was shifted above 0.5 (perfect alternation) and progressively increased with increasing $\alpha$, resulting in continuous transition from trot to gallop only in the direction of left (contralesional) lead sequences. The fore left–right, the contralesional (left) homolateral, and the diagonal phase difference involving the contralesional hind RG all maintained a similar trajectory to their pre-injury counterparts at low $\alpha$ values before the trot-to-gallop bifurcation. On the other hand, the ipsilesional (right) homolateral and diagonal phase differences involving the ipsilesional hind RG were shifted compared with their pre-injury trajectories.

Next, we investigated how injury-affected LPN connectivity and drive to the ipsilesional hind RG contributed to altered gait expression following hemisection injury. We created bifurcation diagrams for simulations in which only the injury-affected LPN connections (*Figure 9A3*) or the drive to the ipsilesional hind RG (*Figure 9A4*) were set to their post-injury recovered values (40% and 90%, respectively), while all other connection and drive weights were left at their intact values.

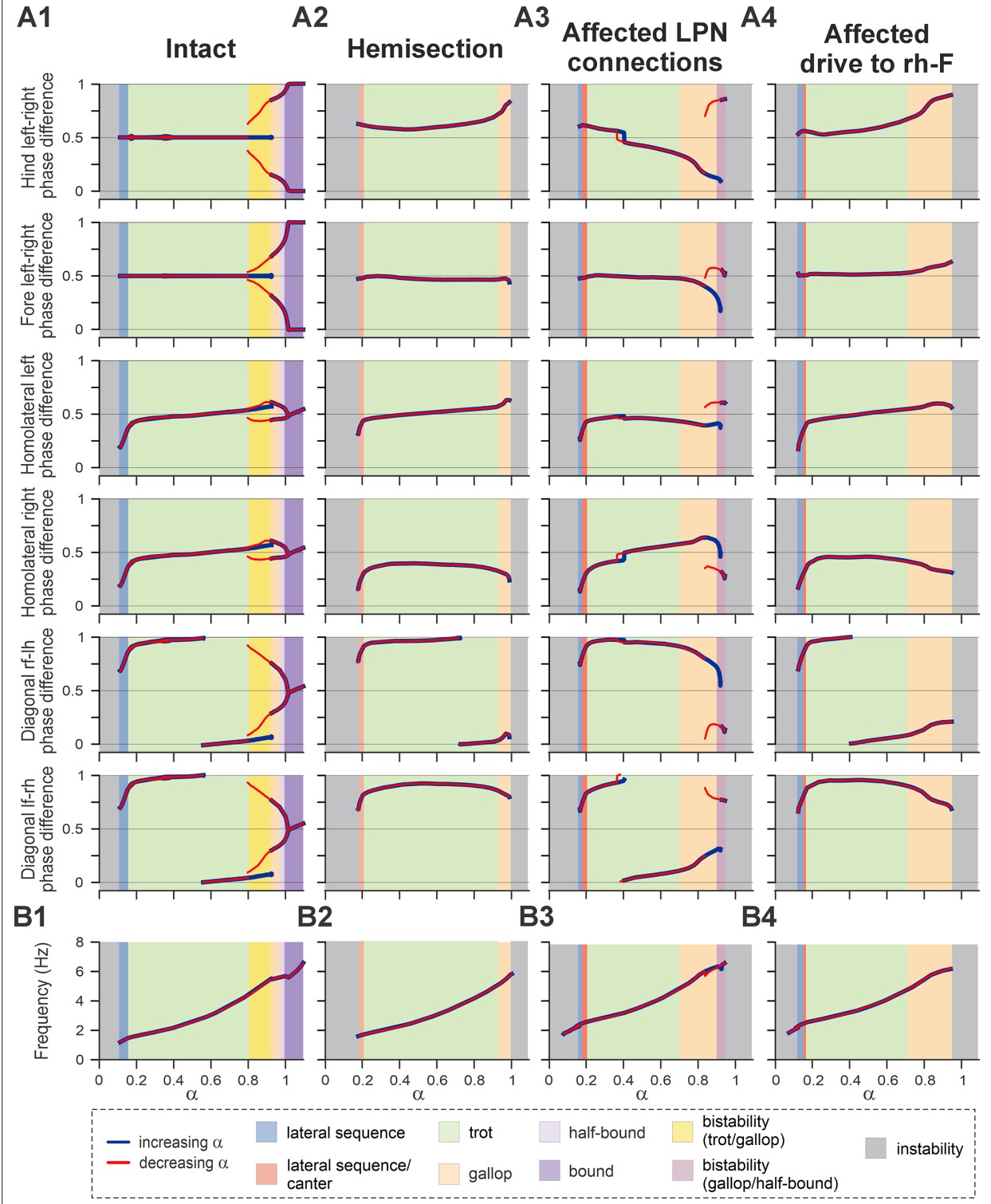

**Figure 9.** Bifurcation diagrams of the intact model (**A1**), following simulated recovery after hemisection (**A2**), and for model versions where only long propriospinal neuron (LPN) connections were affected (40% of the pre-injury values; **A3**) or only brainstem drive to the ipsilesional hind RG was reduced (to 90% of the pre-injury value; **A4**). Diagrams are plotted against the bifurcation parameter α and with reduced noise, $\sigma_{\mathrm{Noise}}$=5 fA. Normalized phase differences of 0.5 correspond to alternation, whereas phase differences of 0 or 1 correspond to synchronization. (**B1**–**B4**) Dependency of locomotor oscillation frequency on parameter α. Blue and red lines indicate stable phase differences or frequency with stepwise increase and decrease of parameter α, respectively. Colored areas indicate the expressed gait. l-, left; r-, right; -f, fore RG/limb; -h, hind RG/limb; RG, rhythm generator.

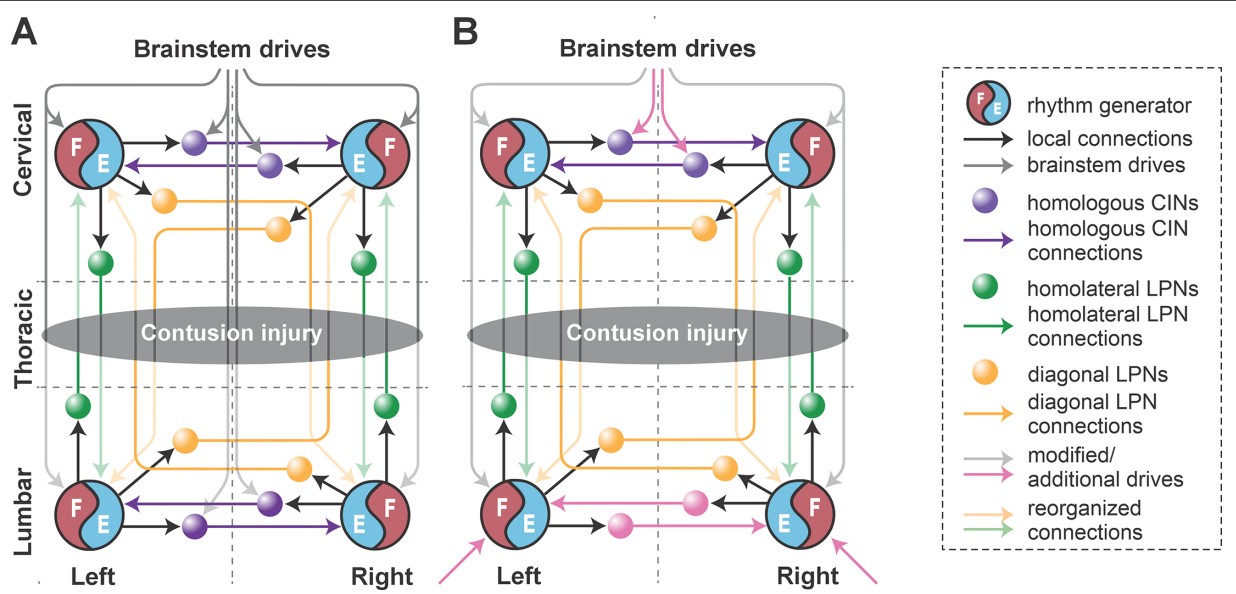

**Figure 10.** Conceptual schematic of the contusion injury (**A**) and following recovery after contusion in the model (**B**). The weights of the long propriospinal neuron (LPN) connections between the cervical and lumbar compartments were significantly reduced. Brainstem drive to the lumbar rhythm generators (RGs) was substituted with additional drives to these RGs (pink arrows at the bottom). Brainstem drives to the cervical RG were adjusted to match oscillation frequency of lumbar RGs (gray arrows at the top). Inhibitory drives to cervical V0v commissural interneurons (CINs) were reduced to secure fore left–right alternation (pink arrows at the top). Commissural pathways in the lumbar compartment were reorganized to secure hind left–right alternation (pink arrows at the bottom).

Interestingly, the lowered connection weights of the injury-affected and partially recovered LPN weights caused a progressively increasing bias of the left–right phase differences with increasing frequency toward ipsilesional (right) lead gallops (*Figure 9A3*) – the opposite direction as observed after hemisection. This is followed by a bistable bifurcation (at ~0.85<α<0.95) toward left-lead sequence gallops.

On the other hand, the slightly (90% of pre-injury value) lowered drive to the ipsilesional hind RG (*Figure 9A4*) caused a left–right asymmetry with a progressive shift away from trot toward gallop and half-bound gallop in the same direction as observed in rats after recovery from hemisection – toward left–lead (contralesional) sequence gaits. Indeed, all six phase differences exhibit trajectories similar to the simulated hemisection (*Figure 9A2*), although with stronger deviations of the left–right fore and hind phase differences from alternation at higher frequencies.

Thus, in the model, injury-induced and partially recovered left–right asymmetries of the LPN connections and descending drive to the ipsilesional hind RG have opposing effects on left–right asymmetry of interlimb coordination and gait expression. The hemisection model is a result of the interaction of both of these influences, but the effect of the lower drive to ipsilesional hind RG compared to the other three RGs dominates and largely causes the observed left–right bias.

## Modeling recovery from thoracic contusion injury

A midline (left–right) symmetrical thoracic contusion injury (*Figure 1C* and *Figure 10A*) partially damages the ascending and descending pathways on both sides of the spinal cord, including supraspinal inputs and long propriospinal connections. However, similar to the hemisection injury, the cervical and lumbar circuitry, including their homologue pathways, remains intact. After a moderate thoracic midline contusion injury, rats regain functional locomotion dominated by left–right alternating gaits (*Danner et al., 2023*), but with even slower maximum speed than after hemisection. They lose the ability to perform non-alternating gaits (gallops and bounds) and adopt atypical gaits not observed in uninjured animals, such as diagonal-sequence and pace. These two novel gaits are characterized by left–right alternation of both the forelimbs and hindlimbs. The diagonal-sequence is a four-beat gait pattern where a hindlimb step is followed by a step of the contralateral (diagonal) forelimb, rather than the ipsilateral forelimb, as is the case in the lateral-sequence. Pace is a two-beat

gait in which the homolateral limbs move in synchrony. Previously, we hypothesized that the post-contusion gait changes are a result of weakened fore–hind coupling together with maintenance of left–right alternation (**Danner et al., 2023**).

We assumed that after symmetrical thoracic contusion injury, the brainstem drive and ascending and descending LPN connections on both sides of the model are significantly damaged (**Figure 1C** and **Figure 10A**). Thus, to model circuit reorganization underlying recovered post-contusion locomotion, we assumed that spared sublesional lumbar circuitry undergoes plastic reorganization and that supraspinal control over the intact supralesional cervical circuitry is altered. Then, by manually tuning model parameters, we looked for possible changes of connection weights in the model that result in reproduction of locomotor function of rats following recovery from contusion, while also keeping the connection weights of the injured pathways low. Note that we did not make any assumption about the proportion of the translesional fibers that remain functionally intact.

This process resulted in the following specific suggestions that we implemented in the model (**Figure 10B**):

1. reduced brainstem drive to the sublesional lumbar RGs was substituted with enhanced afferent input and/or changes in their intrinsic excitability (simulated as additional drives to the hind RGs);
2. descending control over the supralesional cervical circuitry via (tonic) brainstem drive was adjusted so that the oscillation frequency of the cervical RGs matched those of the lumbar RGs (simulated by equal drives to cervical and lumbar RGs) and to secure fore left–right alternation (reduction of inhibitory drives to cervical V0 CINs);
3. commissural pathways in the lumbar compartment were reorganized to secure hind left–right alternation (descending drive to $V0_V$ CINs was removed, and their excitability was altered by an additional constant drive; **Figure 10B**); and
4. finally, connection weights of all inter-enlargement LPN connections were set to 5% of their pre-injury values.

An exemplary bout produced by the post-contusion model is shown in **Figure 11A1**. The model progressively transitioned from a diagonal-sequence gait to trot, then across lateral-sequence to pace and finally back to the diagonal-sequence gait. The example bout of rat post-contusion recovery (**Figure 11A2**) exhibited the same sequence of gait transitions except that it starts and ends with a lateral-sequence gait pattern. Note that fore and hind left–right alternation was maintained across all steps of the bouts, while the homolateral and diagonal phase differences progressively drift in parallel across the full cycle, reaching their approximate initial values at the end of the bouts (**Figure 11A1 and A2**, bottom panels). In both cases, the fore limbs/RGs performed one more step cycle than the hind limbs/RGs (**Figure 11A1 and A2**, upper panels).

Across simulated bouts, the post-contusion model expressed lateral and diagonal-sequence gait patterns, as well as trot and pace (**Figure 5A1**). Like contused rats (**Figure 5A2**; **Danner et al., 2023**), the model did not express non-alternating gaits (gallop, half-bound gallop, and bound) but produced left–right alternating gaits that the intact model did not (including pace and the diagonal-sequence pattern; **Figure 11B1**). The maximal locomotor frequencies were reduced up to approximately ~6 Hz (**Figure 12**), and average gaits of the model exhibited similar extension phase patterns and phase differences to those observed in rats (**Figure 11B1 and B2**). Across the expressed range of frequencies, both fore and hind left–right normalized phase differences remained alternating (around 0.5; **Figure 12A1 and A2**) – which is in contrast to the intact case where fore and hind left–right phase differences progressively transition from alternating to in-phase synchronization with increasing frequency (**Figure 4**). On the other hand, all fore–hind phase differences (both homolateral and diagonal phase differences) were highly variable and distributed across the whole cycle. Frequency dependence of gait expression was largely lost, with all gaits expressed across the full range of frequencies (**Figure 12A1 and A2**). Yet, in the model, diagonal-sequence steps were more frequent at low frequencies and lateral-sequence steps at high frequencies. This is different from contused rats, which expressed these gaits on average at comparable frequencies (**Figure 12B1 and B2**).

Interestingly, gait transition probabilities in the post-contusion model showed a clear pattern (**Figure 5C3 and D3**) and apart from the not-defined category (capturing all other gaits), the expressed gaits were relatively stable: a step of one gait was likely to be followed by another of the same gait (all probabilities 0.74–0.85). Transitions predominantly occurred between neighboring gaits along the

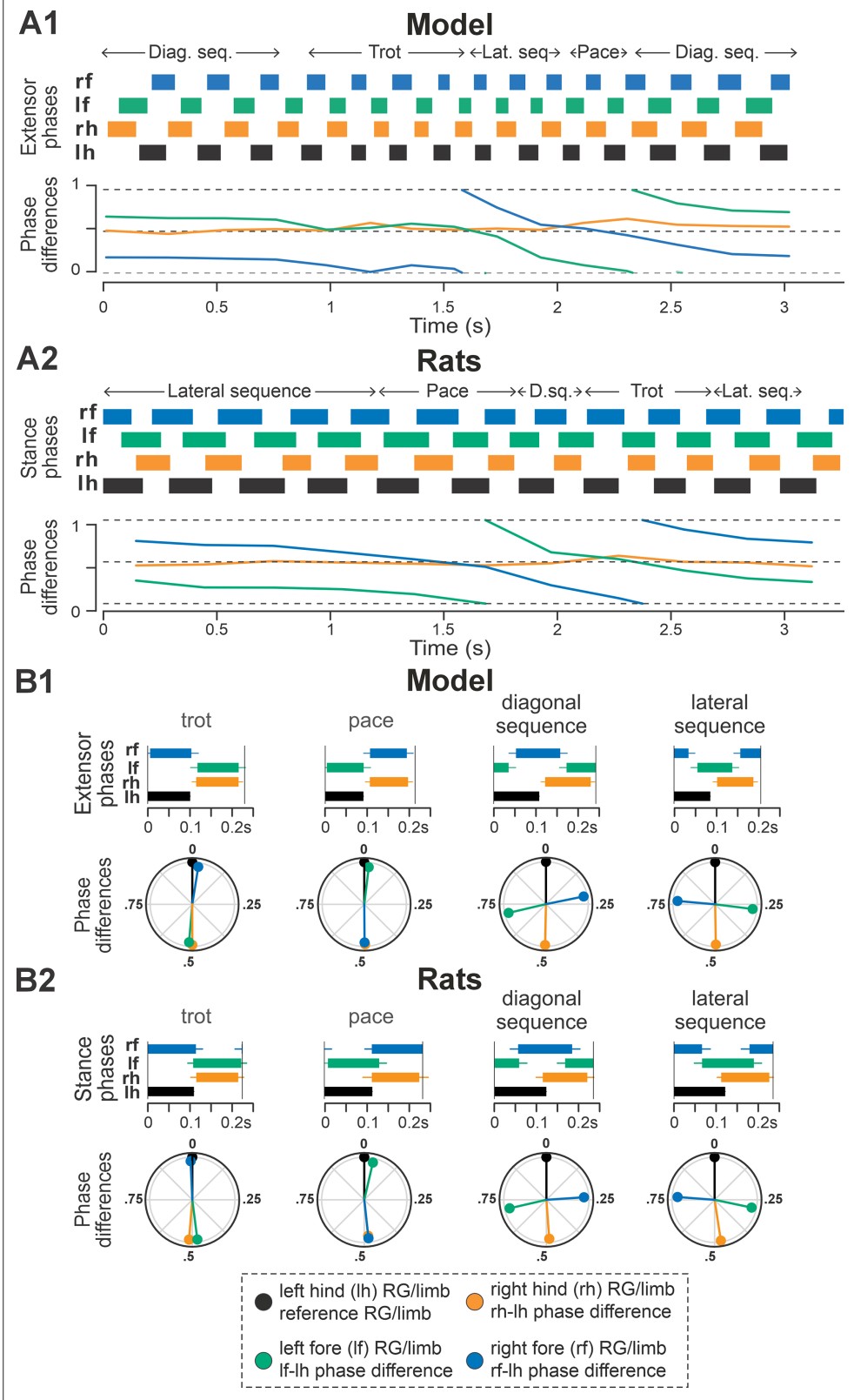

**Figure 11.** Gait expression in the model and rats following recovery after contusion. (**A1**, **A2**) Extensor/stance phases (upper panels) and instantaneous normalized phase differences (bottom panels) of representative bouts for the model (**A1**) and a rat (**A2**). (**B1**, **B2**) Average extensor/stance phases for each gait (upper panels; error bars indicate standard deviations) and circular plots of average normalized phase differences for each gait (bottom

*Figure 11 continued on next page*

*Figure 11 continued*

panels; vector length corresponds to mean resultant length, R) expressed in the post-contusion model (**B1**) and rats (**B2**). Detailed statistical results for rats are reported in *Danner et al., 2023*. D.sq., diag. seq., diagonal-sequence; lat. seq., lateral-sequence; RG, rhythm generator.

---

sequence: lateral-sequence, trot, diagonal-sequence, pace, and back to lateral-sequence (*Figure 5C3 and D3*). The idealized versions of these gaits are evenly distributed along a line (or rather circle) in phase space (defined by the three orthogonal phase differences: hind left–right, homolateral, and diagonal). The line connects trot and pace in both directions (through lateral-sequence in one and diagonal-sequence in the other direction) and represents the only remaining degree of freedom, assuming that fore and hind left–right alternation was maintained. This organization also explains the experimentally observed distribution of phase differences in contused rats (*Danner et al., 2023*).

These results are further complemented by the analysis of phase difference variability (*Figure 5B1 and B2*). Specifically, the mean variability (mean deviation from the moving average) of both fore and hind left–right phase differences was lower compared to the variability of all fore–hind phase differences in both the model and in rats. Comparing variability post-contusion to the pre-injury condition revealed some discrepancies between the model and experimental results. In the model, variability of fore and hind left–right phase differences decreased after contusion, whereas the experimental results showed no significant change. This suggests that the model may have overestimated the strengthening of left–right coupling due to disinhibition of cervical V0$_V$ CINs and reorganization of the lumbar circuitry. However, the mean variability of all fore–hind phase differences (left and right homolateral and diagonal) increased in both the model and experimental results.

Overall, the model supports our previous hypothesis that, following a symmetrical moderate thoracic contusion injury, coupling between the left and right RGs within each enlargement remained at (or recovered back to) pre-injury levels, while the coupling between the lumbar and cervical circuits was significantly reduced. These results suggest that plasticity with the local circuitry linking the left and right RGs within each enlargement is critical for recovery following a symmetrical thoracic contusion injury.

## Sensitivity of hemisection and contusion models to connectivity changes

To evaluate which connectivity changes most strongly shaped speed-dependent locomotor behavior after injury, we performed a local sensitivity analysis of the contusion and hemisection models. This analysis quantified how small variations (in the range of [80%,125%]) in the connection weights of key pathways affect post-injury frequency-dependent gait expression. The outcome measure was the Earth Mover's Distance (EMD; *Rubner et al., 1998*) of each perturbed simulation from the corresponding post-injury baseline model in a seven-dimensional feature space (six interlimb phase differences plus locomotor frequency). Sensitivity was expressed as Sobol indices (*Figure 13*), which quantify the fraction of variance in the EMD attributed to each parameter (*Sobol', 2001*). High indices identify parameters whose perturbations most strongly alter frequency-dependent gait expression (for more details, see Methods).

In the hemisection model, drive to the ipsilesional hind/lumbar flexor RG center was the dominant determinant of locomotor outcomes (*Figure 13A*), with both first- and total-order indices reaching ~0.8–0.9. Varying this drive within the sensitivity range produced large, systematic shifts in phase difference–frequency relationships and gait distributions (*Figure 13—figure supplement 1A and B*), consistent with its high Sobol values. Lumbar commissural interneuron (V0 and V3) connection weights were the only other parameters with appreciable influence (total-order indices ~0.1), and their variation also produced clear changes in phase differences and gait patterns (*Figure 13—figure supplement 1D and E*). In contrast, long propriospinal pathways and cervical CINs showed very low sensitivity indices (≤0.01); e.g., recovered LPN connection weights resulted in outputs nearly indistinguishable from the baseline model (*Figure 13—figure supplement 1A and C*), representative of other low-sensitivity parameters.

A similar pattern was observed in the contusion model. Drive to the sublesional lumbar flexor RG centers dominated the sensitivity profile (total-order 0.96; first-order 0.69; *Figure 13B*), and varying this drive generated robust changes in frequency-dependent distributions of phase differences and

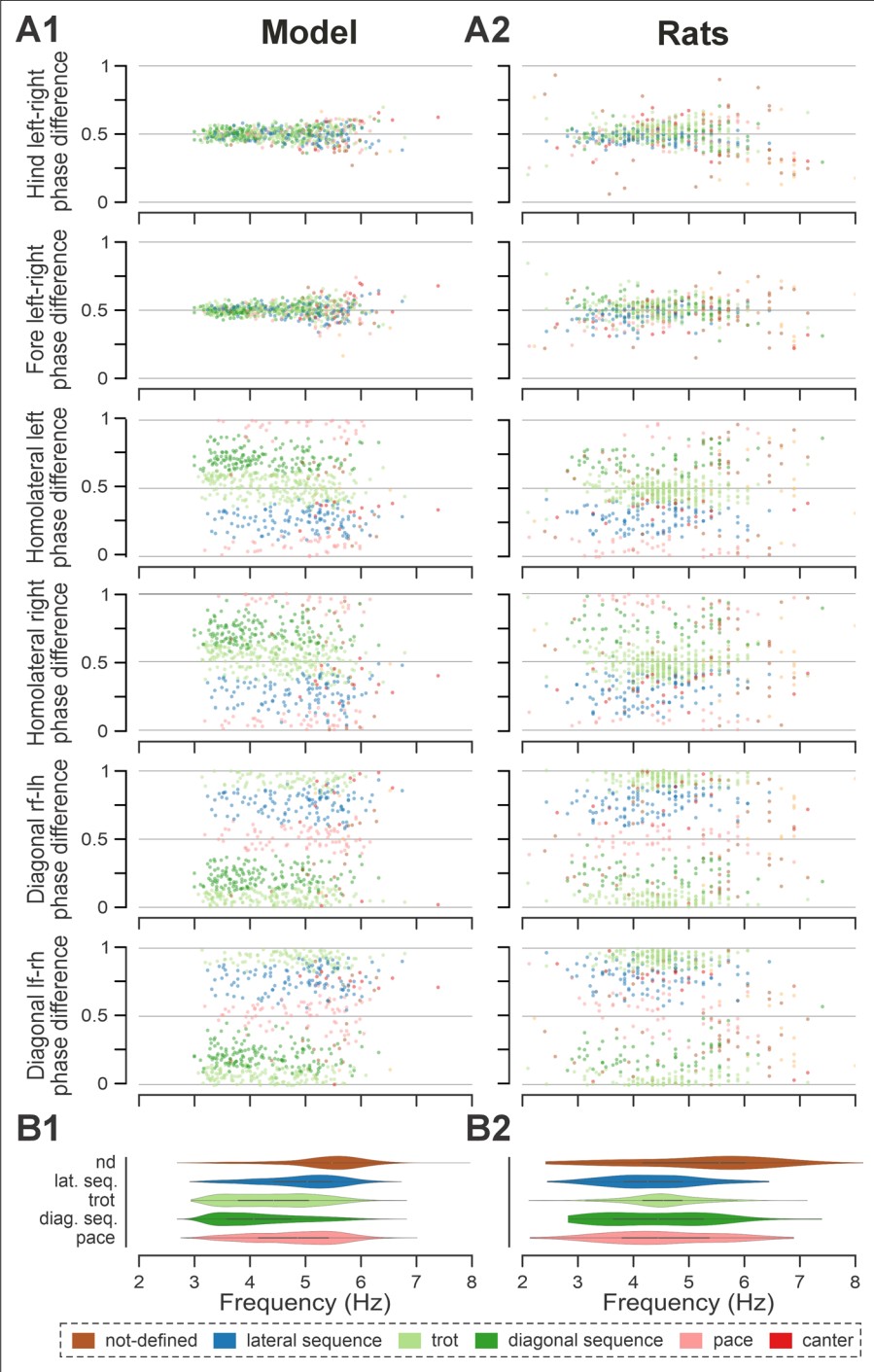

**Figure 12.** Frequency-dependent distribution of normalized phase differences in the model (**A1**) and in rats (**A2**) following recovery after contusion. (**A1**, **A2**) Scatter plots of normalized phase differences are plotted against frequency of locomotor oscillations. Each dot represents one period/step cycle. Gaits are classified for each period/step cycle and color-coded. (**B1**, **B2**) Distribution of gaits vs. locomotor frequency in the model (**B1**) and rats (**B2**). l-, left; r-, right; -f, fore RG/limb; -h, hind RG/limb; RG, rhythm generator.

gaits (*Figure 13—figure supplement 2A and B*). Lumbar V0 and V3 CIN connection weights were the next largest contributors (total-order 0.08–0.22; *Figure 13B*), and their variation produced clear, albeit smaller, distortions in phase relationships and gait expression (*Figure 13—figure supplement 2D and E*). In contrast, cervical CINs and ascending/descending LPN pathways had negligible effects

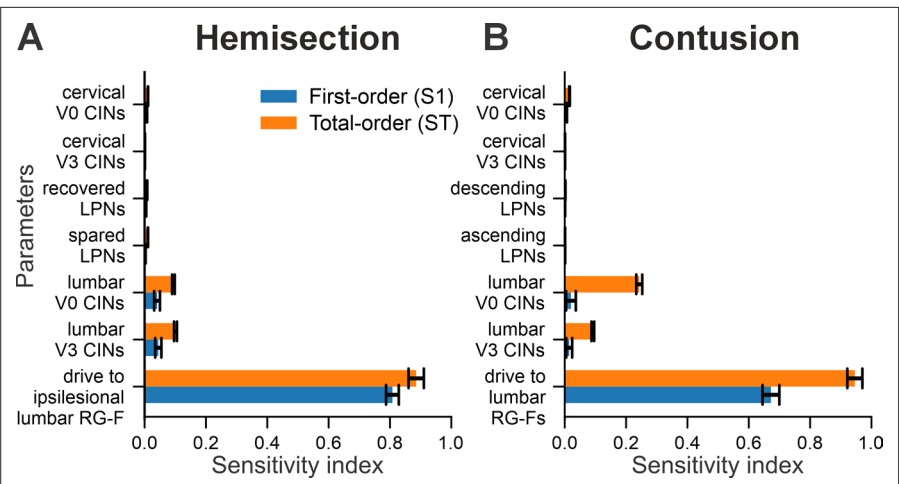

**Figure 13.** Local sensitivity analysis of hemisection (**A**) and contusion (**B**) models. Shown are Sobol first-order (S1) and total-order (ST) sensitivity indices quantifying how small parameter variations (80–125% of baseline) affected locomotor behavior, measured as the Earth Mover's Distance (EMD) across six interlimb phase differences and locomotor frequency. Parameters include drive to the sublesional lumbar flexor rhythm generator centers (RG-F; ipsilesional in hemisection, bilateral in contusion), connection weights of cervical and lumbar V0 and V3 commissural interneuron (CIN), and long propriospinal neuron (LPN) pathways. Error bars indicate 95% confidence intervals.

The online version of this article includes the following figure supplement(s) for figure 13:

**Figure supplement 1.** Effects of parameter variation on model output in the hemisection model.

**Figure supplement 2.** Effects of parameter variation on model output in the contusion model.

**Figure supplement 3.** Predicted Earth Mover's Distance to baseline as a function of lumbar V0/V3 commissural interneuron connectivity.

on locomotor output (total-order indices<0.02), with example simulations of varying ascending LPN weights closely resembling baseline behavior (*Figure 13—figure supplement 2A and C*).

Partial dependence plots further revealed how interactions between lumbar V0 and V3 commissural pathways shape locomotor activity in both hemisection and contusion models (*Figure 13—figure supplement 3*). The large difference between total-order and first-order sensitivity for these pathways (*Figure 13A and B*) indicates that their influence arises mainly through interactions rather than isolated changes to either V0 or V3 pathways. Consistent with this, covarying V0 and V3 together (either increasing or decreasing both within the sensitivity range) kept predicted EMD low, indicating that coordinated scaling preserves overall locomotor behavior. In contrast, disrupting the balance between the two pathways (by changing only one, or by altering them in opposite directions) produced substantially larger deviations from the baseline pattern. These results suggest that the relative balance of V0 and V3 connectivity, rather than their absolute strengths individually, is critical for maintaining post-injury gait expression in the model.

Taken together, the results demonstrate that across both injury models, two classes of parameters substantially influence frequency-dependent gait expression: (1) drive to the sublesional lumbar RGs and (2) lumbar commissural (V0/V3) interneuron connectivity, whose impact depends critically on their coordinated balance. Within the tested perturbation range, changes in LPN pathways and cervical CINs had only minor effects on model output. This emphasizes the central role of restoration of activation of sublesional lumbar rhythm-generating circuits and strengthened commissural pathways in shaping post-injury locomotor patterns.

## Discussion

The brain and spinal cord have a high capacity to adapt and reorganize. Indeed, following incomplete spinal cord injuries, neural plasticity occurs across the breadth of the nervous system (*Filli and Schwab, 2015*). This includes reorganization of supraspinal motor centers, compensatory structural

and functional plasticity of spared descending, propriospinal, and sensory neurons above and below the lesion (*Ballermann and Fouad, 2006*; *Rosenzweig et al., 2010*; *Filli et al., 2014*; *Takeoka et al., 2014*; *Zörner et al., 2014*), local regrowth/sprouting of transected fibers, as well as intrinsic changes to sublesional spinal interneurons and motoneurons (*Barrière et al., 2008*; *Cohen-Adad et al., 2014*; *Takeoka and Arber, 2019*). In this computational modeling study, we have investigated potential reorganization strategies of neural connectivity underlying recovery of locomotor function following thoracic lateral hemisection and moderate midline contusion injuries in rats, representing asymmetrical and symmetrical injuries, respectively, that exhibit roughly comparable functional recovery (*Danner et al., 2023*). The model itself represents neuronal populations resident in the cervical and lumbar enlargements with axons of ascending and descending inter-enlargement propriospinal neurons along with reticulospinal drive to the individual hind and fore RGs. The model reproduced key features of speed-dependent gait expression of rats before and after spinal cord injury. It suggests that recovery after lateral hemisection involves partial functional restoration of descending drive and long propriospinal pathways, whereas recovery following midline contusion relies on reorganization of sublesional lumbar circuitry combined with altered descending control of cervical networks. Across both injuries, sensitivity analyses consistently identified two principal determinants of recovered locomotion: restored excitability of the sublesional lumbar rhythm-generating circuits and appropriate connectivity and balance of lumbar V0/V3 commissural interneuron pathways. These findings highlight the central role of sublesional lumbar rhythm-generating and commissural circuitry in shaping post-injury locomotor function.

Descending drive from the reticular formation controls locomotor speed and gait in intact animals (*Capelli et al., 2017*; *Ausborn et al., 2019*; *Hsu et al., 2023*), and substantial structural and functional plasticity of reticulospinal neurons occurs in parallel with recovery of locomotor function following hemisection injury (*Ballermann and Fouad, 2006*; *Filli et al., 2014*; *Engmann et al., 2020*; *Lemieux et al., 2024*). In our model, drive from the reticular formation controls the oscillation frequency of each RG and modulates the activity of commissural interneurons controlling interlimb coordination. The model suggests that following the asymmetric lateral hemisection injury, the drive to the ipsilesional hind RG recovers to around 90% of its pre-injury strength (*Figure 6B*). The strength of this drive reflects the net excitatory input to the population of flexor RG neurons and cannot be easily interpreted in terms of the number of regenerated axons or synaptic terminals. Thus, while these results suggest substantial reorganization of reticulospinal interactions with the lumbar circuitry, other factors, such as changes in intrinsic neuronal properties, increased prevalence of excitatory connections between constituent neurons, and strengthened afferent input (*Barrière et al., 2008*; *Cohen-Adad et al., 2014*; *Takeoka et al., 2014*; *Takeoka and Arber, 2019*; indicated by the pink arrow to the ipsilesional hind RG in *Figure 6B*) can also affect the excitability of the sublesional RG neurons and help compensate for the reduced descending drive.

Inter-enlargement LPNs couple the cervical circuits controlling the forelimbs with the lumbar circuits controlling the hindlimbs and are essential for interlimb coordination (*Figure 1A* and *Figure 2*; *Edinger, 1896*; *Nathan and Smith, 1959*; *Dutton et al., 2006*; *Ruder et al., 2016*; *Flynn et al., 2017*; *Frigon, 2017*). These neuronal populations are roughly 50% ipsilaterally and 50% contralaterally projecting with the axons of the latter crossing the midline at the level of the cell bodies (*Reed et al., 2006*; *Pocratsky et al., 2020*; *Shepard et al., 2023*). Lateral thoracic hemisection injury severs all these axons unilaterally (*Figure 6A*). Similar to the predicted recovery in descending drive, the model also predicts that partial (functional) recovery of the severed long propriospinal pathways is helpful and possibly necessary for recovery (*Figure 6B*). This recovery was simulated by setting the connection weights of the injured LPN populations and drive connections to a proportion of their pre-injury strengths. Long-distance regeneration of central axons is unlikely to occur in adult animals *Filli and Schwab, 2015*; thus, reorganization of these pathways likely depends on local axon growth and synaptic rewiring onto propriospinal neurons with intact axons, resulting in the formation of detour pathways through the intact hemicord (*Courtine et al., 2008*), as well as compensatory mechanisms of spared reticulospinal neurons and LPNs, including increased innervation of the ipsilesional lumbar hemicord via sprouting (*Ballermann and Fouad, 2006*; *Courtine et al., 2008*; *Takeoka et al., 2014*; *Filli and Schwab, 2015*). Indeed, recovery of locomotor function after hemisection injury has been shown to result in an increased prevalence of LPNs bypassing the injury by crossing the midline twice and of reticulospinal connections to the ipsilesional lumbar cord (*Takeoka et al., 2014*). The work by

*Reed et al., 2006*, uncovered evidence for a small number of descending inter-enlargement neurons with axons that crossed the midline close to the cell body and then again in the lumbar enlargement to terminate ipsilaterally in uninjured animals, suggesting that a descending detour pathway may already exist, even if small. Furthermore, potentiation of ipsilesional muscle responses to optogenetic stimulation of the reticular formation has also been linked to locomotor recovery (*Lemieux et al., 2024*).

Recovery from lateral thoracic hemisection results in distinct left–right asymmetries of locomotor behavior (*Danner et al., 2023*). These asymmetries are most obvious at higher speeds and for non-alternating gaits – animals almost exclusively use the contralesional limb as the lead limb. We investigated the individual contribution of the asymmetries of long propriospinal connections and of the descending drive of the simulated hemisection (*Figure 9A3 and A4*). Interestingly, in the case where only the asymmetrical drive to the lumbar RGs (90% of pre-injury strength to ipsilesional RG) was included and all other connections were set to their pre-injury values, the model (*Figure 9A4*) very closely reproduced frequency-dependent changes of interlimb coordination and gait expression of hemisected animals. On the other hand, the asymmetry introduced by the partially recovered long propriospinal connectivity resulted in gait asymmetries in the opposite direction (ipsilesional limb used as lead limb for non-alternating gaits; *Figure 9A3*). Thus, the asymmetry of the drive to the ipsilesional hindlimb RG compensated for the asymmetry of the long propriospinal connections and was the main contributor to the observed gait asymmetries (*Figures 7–9A2*).

In the case of the midline, symmetrical contusion injury, our model suggests that connection weights of 5% of their pre-injury values for the LPNs are sufficient to reproduce the experimentally observed changes in interlimb coordination and gait expression. These values are much lower than anticipated based on the proportion of axons (white matter) spared by this type of contusion injury, which affects approximately 70–80% of the overall white matter (*Basso et al., 1996*) but presumably a much smaller proportion of the axons in the outermost layers of the ventrolateral funiculus where the long propriospinal tracts are located (*Basso et al., 1996*; *Reed et al., 2006*; *Brown et al., 2024*). Note that we assumed that post-contusion recovery, descending drive to cervical RGs is tightly controlled to match the level of excitability of the lumbar RGs. This assumption was modeled by adjusting the drive so that the oscillation frequency of cervical and lumbar RGs match. In reality, the excitability of the lumbar RGs would be affected by somatosensory afferent input and body biomechanics (*Rossignol et al., 2006*; *Nishikawa et al., 2007*; *Akay et al., 2014*; *Takeoka et al., 2014*; *Takeoka and Arber, 2019*; *Ausborn et al., 2021*; *Kim et al., 2022*). We introduced noise to all neurons to simulate step-to-step variability, but we did not consider afferent feedback or limb/body biomechanics in the model. Thus, we cannot exclude that the long propriospinal connection strengths necessary to reproduce experimental interlimb coordination were underestimated.

Yet, there is experimental support for plasticity resulting in decreased long propriospinal connectivity. Specifically, using viral-based tracing, we recently found that the number of axons/terminals in the cervical enlargement arising from ascending LPN was significantly reduced after recovery from a mild contusion injury (*Brown et al., 2024*), and further that silencing of a proportion of either ascending or descending LPNs after moderate contusion injury actually improves (normalizes) interlimb coordination and other key locomotor indices (*Shepard et al., 2021*; *Shepard et al., 2023*). Thus, LPNs seem to have a disruptive effect on interlimb coordination after injury. One explanation might be that the injury affects different classes of LPNs (ascending vs. descending, homolateral vs. diagonal, left vs. right) differently, leading some to make new connections that are maladaptive to recovery and that (further) reduction of their connectivity via silencing reduces their influence and thereby improves interlimb coordination. In support of this idea, in *Brown et al., 2024*, we also found that ascending propriospinal innervation of segments just caudal to the thoracic injury was increased in the same animals where input to C6 was decreased, all relative to uninjured animals. Thus, unguided plasticity of LPNs post-injury seems to have a disruptive effect on interlimb coordination.

Silencing of ascending or descending LPNs in intact mice or rats has the opposite effect compared to following contusion; it disrupts interlimb coordination in a speed- and context-dependent manner, causing transient periods of stepping with disordered left–right coordination (*Ruder et al., 2016*; *Pocratsky et al., 2020*). Our previous models attributed this effect to the excitatory diagonally projecting LPNs (ascending V3 LPNs, descending V0$_V$ LPNs), which ensure diagonal synchronization for trot but also promote left–right alternation because of the tightly controlled homolateral fore–hind alternation by ipsilaterally projecting LPNs (*Danner et al., 2017*; *Ausborn et al., 2021*; *Zhang*

*et al., 2022*). Thus, the influence of long propriospinal interneurons promoting left–right alternation of cervical and lumbar RGs is reduced following contusion injury. Indeed, somewhat more severe contusion injuries that remove more than 80% of white matter result in an increased prevalence of disrupted left–right alternation and other persistent problems such as dorsal stepping and the loss of the 1:1 forelimb/hindlimb relationship during stepping (*Shepard et al., 2021*; *Shepard et al., 2023*). Yet, recovery from moderate contusion injuries – where plantar stepping and a 1:1 forelimb-to-hindlimb ratio are recovered, as modeled here (*Danner et al., 2023*) – results in a well-controlled left–right alternation of both fore and hindlimbs. Even animals that show the poorest recovery regain stepping, and the number of steps with appropriate left–right alternation significantly exceeds the number of disrupted steps (*Shepard et al., 2021*; *Shepard et al., 2023*). Together, these findings suggest that reorganization and/or altered modulation of intra-enlargement commissural interneurons occurs during recovery to strengthen left–right alternating influences. In the model, this was implemented by reduced descending inhibition of cervical V0 commissural interneurons and altered excitability of lumbar V0 commissural interneurons. This specific implementation reflects one possible scenario, as assumed by the model, where V0 commissural interneuron excitability is regulated by descending inhibition. Yet, descending excitation (*Danner et al., 2016*) or descending neuromodulatory influences – such as serotonergic, noradrenergic, or dopaminergic input – may also contribute (*Butt and Kiehn, 2003*; *Jordan et al., 2008*; *Abbinanti and Harris-Warrick, 2012*; *Abbinanti et al., 2012*; *Perrier and Cotel, 2015*). In addition, local excitatory V2a (*Crone et al., 2009*) and local inhibitory interneurons contribute to left–right coordination (*Kjaerulff and Kiehn, 1997*; *Butt and Kiehn, 2003*; *Shevtsova et al., 2015*; *Danner et al., 2016*; *Danner et al., 2017*; *Kiehn, 2016*) and may be modulated by descending input, making them potential targets for reorganization following injury.

## Model limitations and future direction

Our model has several limitations. Most importantly, it does not incorporate afferent feedback or body and limb biomechanics, both of which play a major role in both pre- and post-injury locomotion and can affect interlimb coordination (*Forssberg et al., 1980*; *Rossignol et al., 2006*; *Nishikawa et al., 2007*; *Aoi et al., 2013*; *Akay et al., 2014*; *Takeoka et al., 2014*; *Takeoka and Arber, 2019*; *Kim et al., 2022*; *Molkov et al., 2024*). Some of the connections in our model might indeed be mediated by afferent feedback: e.g., ascending LPNs receive afferent feedback (*Lloyd and Mcintyre, 1948*; *Flynn et al., 2017*; *Frigon, 2017*; *Laflamme et al., 2023*), and commissural interneurons integrate afferent input from various modalities (*Edgley et al., 2003*; *Jankowska, 2008*; *Jankowska et al., 2009*; *Laflamme et al., 2023*). Furthermore, we did not explicitly model reorganization of afferent feedback connectivity (*Takeoka et al., 2014*; *Takeoka and Arber, 2019*). In reality, restoration of sublesional RG activation may arise not only from partial recovery of descending pathways but also from enhanced somatosensory feedback that partially compensates for lost drive (*Forssberg et al., 1980*; *Lovely et al., 1986*; *Barbeau and Rossignol, 1987*; *Takeoka et al., 2014*; *Frigon, 2017*; *Takeoka and Arber, 2019*); in the present model, these contributions are lumped together and represented only as changes in effective drive to the RGs. Moreover, we note that the reorganizations presented here represent solutions that are sufficient to reproduce experimentally observed post-injury gait expression and alternative circuit configurations may likewise support recovery. Finally, because our modeling framework identified circuit reorganizations that successfully reproduced the observed post-injury speed-dependent gait expression, it primarily reveals reorganizations that support functional recovery and does not capture potentially maladaptive changes that impair locomotor performance.

Future work will require neuromechanical models that explicitly incorporate phasic afferent feedback, its reorganization after injury, and its interaction with body and limb biomechanics. Experimentally, a systematic characterization of post-injury reorganization in lumbar circuits and their interacting pathways (including supraspinal, long propriospinal, commissural, and afferent inputs, with particular emphasis on those that restore activation of sublesional RGs) is needed, as these elements emerged as central to recovered function in our model. Constraining and validating extended models with such data will be critical for distinguishing circuit reorganizations that promote locomotor recovery from those that are maladaptive and limit it.

## Conclusion

In conclusion, challenging animals to perform overground locomotion across the full range of speeds before and after two distinct spinal cord injury models revealed injury-specific deficits in interlimb coordination and gait expression (*Danner et al., 2023*). These deficits differed between the two injury models and extended beyond those observed at walk–trot speeds only. By reproducing these behavioral changes in speed-dependent gait expression in a computational model of spinal loco-motor circuitry, we identified potential principles of circuit reorganization underlying post-injury loco-motor recovery following asymmetrical lateral hemisection and symmetrical contusion injuries at the low-thoracic level.

The model qualitatively recapitulated the *in vivo* locomotor characteristics observed after recovery from lateral hemisection when the ipsilesional descending drive and severed LPNs were partially recovered, suggesting that the plastic changes responsible for recovery involved new and/or strengthened pathways that bypass the lesion to drive the hindlimb RGs. In contrast, the model changes that effectively recapitulated locomotor recovery after a symmetrical midline contusion injury involved strengthening of the local left–right circuitry within each enlargement. These differences suggest that injury symmetry partly dictates the location and type of plasticity supporting recovery.

Despite these contrasts, the analyses converge on a shared requirement for recovered locomotion: reactivation of sublesional lumbar rhythm-generating circuits together with balanced lumbar commissural connectivity. Thus, asymmetric and symmetric injuries may rely on different pathways to achieve a common end point, but restored excitability of lumbar RGs and effective commissural coordination consistently emerge as key determinants of post-injury gait expression.

Accordingly, clinical approaches – including neuromodulation and strategies to enhance regeneration or sprouting – may be more effective when aimed at restoring sublesional lumbar RG and commissural function in a manner tailored to injury symmetry. Finally, we anticipate that further development of such computational approaches will eventually support the creation of individualized models to better characterize injury profiles and assist in the development of therapeutic strategies.

## Methods

### Experimental data

In this modeling study, we used previously published experimental data on gait characteristics of overground locomotion within the full range of speeds in intact rats and following their recovery from either a mild-moderate contusion or a lateral hemisection spinal cord injury (*Danner et al., 2023*). In those studies, all experiments were performed in accordance with the Public Health Service Policy on Humane Care and Use of Laboratory Animals, and with the approval of the Institutional Animal Care and Use Committee (IACUC, protocol number 19644) at the University of Louisville. No new animals were used in the current study. Locomotion was assessed in a 3 m long Plexiglas tank where the walking surface was coated with Sylgard to provide increased grip. A group of 17 adult female Sprague–Dawley rats (225–250 g) was trained, using food treats, to traverse the length of the tank. Recordings were made using four high-speed (200 Hz) video cameras placed to capture the ventral view of the animal and analyzed using custom software. Then rats were divided into two groups and given either lateral hemisection (nine rats) or midline contusion injuries (eight rats; 12.5 g cm) at the T10 level of the spinal cord. The animals were allowed to recover for at least 4 weeks and were reintroduced to the long-tank setup and trained again. At this time, their locomotor recovery plateaued (contusion: 15.7±1.9, hemisection 18.3±0.8 BBB Open Field Locomotor Scale). The post-recovery recordings were analyzed for each type of injury and compared to those for the intact animals. For more details, see *Danner et al., 2023*.

### Modeling methods

Our model represents a network of interacting neural populations. Each population was simulated as an activity-based (firing rate) neuronal population model. In this description, the voltage variable $V$ represents the average membrane potential of the population, and the output function $f(V)$ transforms $V$ to the integrated population activity or normalized average firing rate at the corresponding voltage $V$ (*Ermentrout, 1994*). This description was used in our previous modeling studies (*Rubin et al., 2009*; *Danner et al., 2016*; *Danner et al., 2017*; *Ausborn et al., 2019*; *Latash et al., 2020*;

*Rybak et al., 2024*) and allows for inclusion of explicitly represented ionic currents, particularly, the leakage current $I_L$ and the persistent (slowly inactivating) sodium current, $I_{NaP}$ (*Rubin et al., 2009*). The latter current was proposed to be responsible for generating rhythmic activity in the spinal cord (*Rybak et al., 2006*; *McCrea and Rybak, 2007*; *Tazerart et al., 2007*; *Tazerart et al., 2008*; *Brocard et al., 2013*) and was included in the description of flexor and extensor RG centers.

The variable $V$ for flexor and extensor centers obeyed the differential equation:

$$C \cdot \frac{dV}{dt} = -I_{NaP} - I_L - I_{SynE} - I_{SynI} - I_{Noise}. \tag{1}$$

All other populations did not include $I_{NaP}$, and in those populations, $V$ was described as:

$$C \cdot \frac{dV}{dt} = -I_L - I_{SynE} - I_{SynI} - I_{Noise}, \tag{2}$$

where $C$ is the membrane capacitance, $I_{NaP}$ is the persistent sodium current, $I_L$ is the leakage current, $I_{SynE}$ and $I_{SynI}$ excitatory and inhibitory synaptic currents, respectively, and $I_{Noise}$ a noisy current.

The leakage current was described as:

$$I_L = g_L \cdot (V - E_L), \tag{3}$$

where $g_L$ is the leakage conductance and $E_L$ represents the leakage reversal potential.

The persistent sodium current in the flexor and extensor half-centers was described as:

$$I_{NaP} = \bar{g}_{NaP} \cdot m(V) \cdot h(V) \cdot (V - E_{Na}), \tag{4}$$

where $\bar{g}_{NaP}$ is the $I_{NaP}$ maximal conductance, $E_{Na}$ is the sodium reversal potential, and $m(V)$ and $h(V)$ are voltage-dependent activation and (slow) inactivation variables of $I_{NaP}$, respectively. $m(V)$ was instantaneous and its steady state was described as

$$m(V) = m_\infty(V) = \left\{ 1 + \exp\left[ \frac{(V - V_{1/2,m})}{k_m} \right] \right\}^{-1}, \tag{5}$$

and the slow $I_{NaP}$ inactivation, $h(V)$, was modeled by the differential equation

$$\tau_h(V) \cdot \frac{dh}{dt} = h_\infty(V) - h, \tag{6}$$

$$h_\infty(V) = \left\{ 1 + \exp\left[ \frac{(V - V_{1/2,h})}{k_h} \right] \right\}^{-1}, \tag{7}$$

$$\tau_h(V) = \tau_0 + (\tau_{max} - \tau_0) / \cosh\left[ \frac{(V - V_{1/2,\tau})}{k_\tau} \right], \tag{8}$$

where $h_\infty(V)$ is the inactivation steady state and $\tau_h(V)$ the inactivation time constant.

In *Equations 5–8*, $V_{1/2}$ and $k$ represent half-voltage and slope of the corresponding variables ($m$, $h$, and $\tau$); $\tau_0$ and $\tau_{max}$ are the baseline and maximum of inactivation time constant $\tau_h$, respectively.

In *Equations 1 and 2*, excitatory and inhibitory synaptic currents ($I_{SynE}$ and $I_{SynI}$, respectively) for population $i$ were described as:

$$I_{SynE,i} = g_{SynI} \cdot \left\{ \sum_j [S(w_{ji}) \cdot f(V_j)] + D_{E,i} \right\} \cdot (V_i - E_{SynE}); \tag{9}$$

$$I_{SynI,i} = g_{SynI} \cdot \left\{ \sum_j [S(-w_{ji}) \cdot f(V_j)] + D_{I,i} \right\} \cdot (V_i - E_{SynI}), \tag{10}$$

where $g_{SynE}$ and $g_{SynI}$ are synaptic conductances and $E_{SynE}$ and $E_{SynI}$ are the reversal potentials of the excitatory and inhibitory synapses, respectively; $w_{ji}$ is the synaptic weight from population $j$ to population $i$ ($w_{ji} > 0$ for excitatory connections and $w_{ji} < 0$ for inhibitory connections); function $S$ is described as follows:

$$S(x) = \begin{cases} x, & \text{if } x \geq 0 \\ 0, & \text{if } x < 0 \end{cases} . \tag{11}$$

The output function $f(V)$ $(0 \leq f(V) \leq 1)$ was defined as

$$f(V) = \begin{cases} 0, & \text{if } V < V_{thr} \\ \dfrac{V - V_{\text{thr}}}{V_{\text{max}} - V_{\text{thr}}}, & \text{if } V_{thr} \leq V < V_{\text{max}} \\ 1, & \text{if } V \geq V_{\text{max}} \end{cases} . \tag{12}$$

Excitatory $D_{\text{E},i}$ and inhibitory $D_{\text{I},i}$ drives to population $i$ were modeled as a linear function of the free parameter $\alpha$:

$$D_{\{E,I\},i} = d_{\{E,I\},i} \cdot \alpha + b_{\{E,I\},i}, \tag{13}$$

where $d_{\{E,I\},i}$ is the slope and $b_{\{E,I\},i}$ is the intercept; the free parameter $\alpha$ represents the strength of the brainstem drive.

The noisy current $I_{\text{Noise},i}$ for each population $i$ was described as an Ornstein–Uhlenbeck process

$$dI_{\text{Noise},i} = -\frac{I_{\text{Noise},i}}{\tau_{\text{Noise}}} dt + \sigma_{\text{Noise}} \cdot \sqrt{\frac{2}{\tau_{\text{Noise}}}} dW_i, \tag{14}$$

where $\tau_{\text{Noise}}$ is the time constant, $\sigma_{\text{Noise}}$ the standard deviation of the noise, and $W_i$ a Wiener process. It was numerically simulated using the finite difference formula

$$I_{\text{Noise},i}(t + \Delta t) = I_{\text{Noise},i}(t) - \frac{I_{\text{Noise},i}(t)}{\tau_{\text{Noise}}} \Delta t + \sigma_{\text{Noise}} \sqrt{\frac{2 \Delta t}{\tau_{\text{Noise}}}} \xi_i, \tag{15}$$

where $\xi_i \sim \mathfrak{N}(0,1)$ is a normally distributed random number with a variance of 1 and $\Delta t$ the time step (1 ms).

## Model parameters

The general neuron parameters were adapted from our previous models (**Danner et al., 2016**; **Danner et al., 2017**) to provide the frequency range of locomotor activity characteristic for rats. The following values of parameters were used: $C = 10\,\text{pF}$; $g_L = 4.5\,\text{nS}$ for RG centers and $g_L = 2.8\,\text{nS}$ for all other neurons; $\bar{g}_{\text{NaP}} = 4.5\,\text{nS}$; $g_{\text{synE}} = g_{\text{synI}} = 10\,\text{nS}$; $E_L = -62.5\,\text{mV}$ for RG centers, $E_L = -60\,\text{mV}$ for all other populations; $E_{\text{Na}} = 50.0\,\text{mV}$; $E_{\text{EsynE}} = -10\,\text{mV}$; $E_{\text{EsynI}} = -75\,\text{mV}$; $V_{\text{thr}} = -50\,\text{mV}$; $V_{\text{max}} = 0\,\text{mV}$; $V_{1/2,m} = -40.0\,\text{mV}$; $k_m = -6\text{mV}$; $V_{1/2,h} = -45.0\,\text{mV}$; $k_h = 4\text{mV}$; $\tau_{\text{max}} = 400\,\text{ms}$; $\tau_0 = 150\,\text{ms}$; $V_{1/2,\tau} = -35\,\text{mV}$.

The following drive parameters were used for the intact model: $d_E = 0.0$ and $b_E = 0.1$ for th e extensor centers; $d_E = 0.1$ and $b_E = 0.0$ for the flexor centers; $d_I = 0.75$ and $b_I = 0.0$ for homologous (within a girdle) $V0_D$ CINs in both cervical and lumbar compartments; $d_I = 1.5$ and $b_I = 0.0$ for diagonal $V0_D$ LPNs (d$V0_D$); $d_I = 0.25$ and $b_I = 0.0$ for the homologous cervical (fore) $V0_V$ CINs; $d_I = 0.15$ and $b_I = 0.0$ for the homologous lumbar (hind) $V0_V$ CINs. Connection weights for the intact model are listed in **Table 1**.

To simulate rat behavior after recovery from right hemisection, drive to the ipsilesional lumbar flexor center was reduced by 10%, inhibitory drives to both cervical and lumbar $V0_V$ CINs were reduced by 50%, and connection weights of LPN connections affected by the injury were reduced to 40% of their pre-injury values (see **Table 2** for a comprehensive list of differences to the pre-injury case).

To simulate rat behavior after recovery from contusion, all weights of LPN connection were reduced by 95%. In addition, the increasing inhibitory drives to lumbar $V0_D$ and $V0_V$ CINs were eliminated and replaced by constant drives (see **Table 3** for a comprehensive list of differences to the pre-injury case).

## Computer simulations

The set of differential equations was solved using the odeint (**Ahnert et al., 2011**) implementation of the Cash–Karp fifth-order Runge–Kutta method (**Cash and Karp, 1990**) of the Boost C++ library (version 1.88.0). The custom C++ code was compiled (with clang-1700.0.13.3 for MAC OS 15.3.1) as

**Table 1.** Connection weights in the intact model.

| Source | Target ($w_{ij}$) |
| --- | --- |
| Within cervical and lumbar circuits | |
| RG-F | i-InF (0.4), i-V0$_D$ (0.7), i-V2a (1) |
| RG-E | i-InE (0.4), i-V3-E (0.35), i-Sh2 (0.5) |
| IniF | i-RG-E (–1) |
| IniE | i-RG-F (–0.1) |
| V2a | i-V0$_V$ (1) |
| V0$_V$ | c-Ini (0.6) |
| V0$_D$ | c-RG-F (–0.07) |
| V3-E | c-RG-E (0.02) |
| Within cervical circuits | |
| RG-F | i-dLPNi (0.7), i-dV0$_D$ (0.5), i-dV2a (0.5) |
| Ini | i-RG-F (–0.0375) |
| dV2a | i-dV0$_V$ (0.9) |
| Within lumbar circuits | |
| RG-F | i-V3-F (0.4), i-aV3 (0.3) |
| Ini | i-RG-F (–0.075) |
| V3-F | c-RG-F (0.03) |
| V3-E | c-InE1 (1) |
| InE1 | c-RG-E (–0.045) |
| Between cervical and lumbar circuits | |
| dSh2 | ih-RG-F (0.005) |
| aSh2 | if-RG-F (0.04) |
| dLPNi | ih-RG-F (–0.01) |
| dV0$_D$ | ch-RG-F (–0.075) |
| dV0$_V$ | ch-RG-F (0.02) |
| aV3 | cf-RG-F (0.065) |

i-, ipsilateral; c-, contralateral; f-, fore; h-, hind.

a Python package. Python 3.11.5 (Python Software Foundation, Wilmington, DE, USA) was used to run simulations and further analyze the results. The neural network simulation package is available at https://github.com/SimonDanner/CPGNetworkSimulator (copy archived at *Danner, 2025*).

## Data analysis

Activities of the flexor and extensor centers were used to assess the model behavior. Each RG was considered as being in flexion when output $f(V)$ of its flexor center was greater than or equal to 0.1; otherwise, it was considered as being in extension. The midextension time was defined as the halfway between onset and offset of extension. The locomotor period (step) was defined as the duration between two consecutive midextension time points; the frequency as the reciprocal of the period; extension and flexion phase durations as the duration between on- and offset of extension and flexion, respectively. Normalized phase differences were calculated as the delay between the midextension time points of a pair of RGs divided by the locomotor period. Six normalized phase differences were calculated: hind left–right (right hind–left hind), fore left–right (right fore–left fore), left homolateral

**Table 2.** Hemisection: differences to pre-injury model.

**Connection weights**

| Source | Target $\left(w_{\mathbf{intact}} \rightarrow w_{\mathbf{hemisection}}\right)$ |
|---|---|
| hl-aV3 | fl-RG-F (0.065 → 0.026) |
| hr-aSh2 | fr-RG-F (0.04 → 0.016) |
| fr-dSh2 | hr-RG-F (0.005 → 0.002) |
| fr-dLPNi | hr-RG-F (–0.01 → –0.004) |
| fl-dV0$_\mathrm{D}$ | hl-RG-F (–0.075 → –0.03) |
| fl-dV0$_\mathrm{V}$ | hl-RG-F (0.02 → 0.008) |

**Drive parameters**

| Target | $d_{\{\mathbf{E,I}\},\mathbf{intact}} \rightarrow d_{\{\mathbf{E,I}\},\mathbf{hemisection}}$ |
|---|---|
| rh-RG-F | $d_E = 0.1 \rightarrow d_E = 0.009$ |
| h-V0$_\mathrm{V}$ | $d_E = 0.15 \rightarrow d_I = 0.075$ |
| f-V0$_\mathrm{V}$ | $d_I = 0.25 \rightarrow d_I = 0.125$ |

f-, fore; h-, hind; l-, left; r-, right.

(left fore–left hind), right homolateral (right fore–right hind), and two diagonal (right fore–left hind and left fore–right hind).

## Gait classification

Gaits for each step cycle were operationally defined based on phase differences, similarly to the method used in *Danner et al., 2023*. Each step was represented as a point $\Phi$ in the phase space consisting of the three (non-normalized) phase differences with the left hind RG as a reference. Then, the distance to all idealized one-, two-, three-, and four-beat gaits ($\Psi_g$; *Table 4*) was calculated

**Table 3.** Contusion: differences to pre-injury model.

**Connection weights**

| Source | Target$\left(w_{\mathbf{intact}} \rightarrow w_{\mathbf{contusion}}\right)$ |
|---|---|
| h-aV3 | cf-RG-F (0.065 → 0.00325) |
| h-aSh2 | if-RG-F (0.04 → 0.002) |
| f-dSh2 | ih-RG-F (0.005 → 0.00025) |
| f-dLPNi | ih-RG-F (–0.01 → –0.0005) |
| f-dV0$_\mathrm{D}$ | ch-RG-F (–0.075 → –0.00375) |
| f-dV0$_\mathrm{V}$ | ch-RG-F (0.02 → 0.001) |

**Drive parameters**

| Target | $d/b_{\{\mathbf{E,I}\},\mathbf{intact}} \rightarrow d/b_{\{\mathbf{E,I}\},\mathbf{contusion}}$ |
|---|---|
| h-V0$_\mathrm{D}$ | $d_I = 0.15 \rightarrow 1.d_I = 0.0; b_I = 0.0 \rightarrow b_I = 0.2$ |
| h-V0$_\mathrm{V}$ | $d_I = 0.25 \rightarrow 1.d_I = 0.0; b_I = 0.0 \rightarrow b_I = 0.03$ |

i-, ipsilateral; c-, contralateral; f-, fore; h-, hind.

**Table 4.** Idealized gaits.

| | Normalized phase differences | | | Gait |
|---|---|---|---|---|
| | LR | HL | diag. | |
| One-beat | 0 | 0 | 0 | Pronk |
| | 1/2 | 1/2 | 0 | Trot |
| | 0 | 1/2 | 1/2 | Bound |
| | 0 | 2/3 | 2/3 | Bound |
| Two-beat | 1/2 | 0 | 1/2 | Pace |
| | 0 | 1/3 | 2/3 | Half-bound |
| | 0 | 2/3 | 1/3 | Half-bound |
| | 2/3 | 1/3 | 0 | Canter |
| | 1/3 | 1/3 | 2/3 | Canter |
| | 1/3 | 2/3 | 0 | Other |
| | 2/3 | 2/3 | 1/3 | Other |
| | 1/3 | 2/3 | 2/3 | Other |
| | 2/3 | 1/3 | 1/3 | Other |
| | 1/3 | 0 | 2/3 | Other |
| | 2/3 | 0 | 1/3 | Other |
| | 1/3 | 2/3 | 1/3 | Other |
| Three-beat | 2/3 | 1/3 | 2/3 | Other |
| | 3/4 | 1/4 | 2/4 | Rotary gallop |
| | 1/4 | 3/4 | 2/4 | Rotary gallop |
| | 3/4 | 2/4 | 1/4 | Transverse gallop |
| | 1/4 | 2/4 | 3/4 | Transverse gallop |
| | 2/4 | 1/4 | 3/4 | Lateral-sequence |
| Four-beat | 2/4 | 3/4 | 1/4 | Diagonal-sequence |

LR: hind left–right normalized phase difference, HL: homolateral normalized phase difference; diag.: diagonal normalized phase difference. Adapted from *Danner et al., 2023*.

$$d\left(\mathbf{\Phi}, \mathbf{\Psi}_{\mathrm{g}}\right) = \sqrt{\sum_{j=\{\mathrm{LR,HL,DIAG}\}} \arg\left[e^{i \cdot \left(\phi_j - \psi_j\right)}\right]^2}$$

(16)

and the step cycle was assigned to belong to the gait $\mathbf{\Psi}_{\mathrm{g}}$ to which it was closest. The duty factor was not considered for gait classification. Gait transitions were analyzed on a step-by-step basis (*Danner et al., 2023*).

## Analysis of model performance

To model step-to-step variability and gait transitions comparable with locomotor bouts of rats in the long-tank (*Danner et al., 2023*), simulations with increased noisy currents ($\sigma_{\mathrm{Noise}} = 1.1$ pA) were performed in all three conditions: intact, post-hemisection, and post-contusion. In these simulations, $\alpha$ was gradually ramped up from 0.55 to 1.05 for the intact condition, or to 1.0 for hemisection and contusion conditions, and then ramped back down to 0.55. Each phase of the ramp lasted 8 s, and the entire ramp cycle was repeated 200 times. Variability of phase differences was quantified as

the deviation of each phase difference from its circular moving average (cutoff frequency 0.125 step cycles) averaged across ramp cycles (*Danner et al., 2023*).

To identify stable solutions of the model, bifurcation diagrams (*Danner et al., 2017*) were built for the six normalized phase differences. For this purpose, $\alpha$ was incrementally increased from 0.075 to 1.1, and then decreased back to 0.075, using a step size of 0.001. At each step, the simulation was run for 10 s, then if the normalized circular standard deviation of the five last step cycles of each phase difference was smaller than 0.005, the final state was used as the initial condition for the next step; otherwise, the same procedure was repeated. If the normalized standard deviation was not met by 100 s of simulation, the model was considered not stable at this specific $\alpha$ value. Otherwise, the mean phase differences were regarded as the stable solutions. Initial conditions were randomized, and multiple runs were evaluated. To ensure that the model does not remain on an unstable trajectory, these simulations were performed with a weak noisy current ($\sigma_{\mathrm{Noise}} = 5$ fA).

## Sensitivity analysis

To identify which connectivity changes most strongly influence post-injury gait, we performed a local sensitivity analysis of the injured (contusion and hemisection) models. For each model, we quantified how parameter variation altered locomotor behavior relative to the baseline parameter set. Simulations with noise were performed for each parameter perturbation, as described above but with 25 ramp cycles.

As a summary outcome, we computed the EMD (*Rubner et al., 1998*) between the baseline simulation and each perturbed simulation in a seven-dimensional feature space defined by the six interlimb phase differences and locomotor frequency. EMD measures the minimal 'work' required to transform one multivariate pattern into another and thus provides a single, interpretable metric of overall locomotor pattern disruption. Distances were computed as the sum of circular (torus) distances for the six interlimb phase differences combined with the Euclidean distance for frequency.

Model parameters were perturbed multiplicatively on the logarithmic scale $p' = p \cdot e^x$, $x \sim \mathrm{Uniform}(-\ln r, \ln r)$, where $p$ is the baseline parameter value, and $p'$ the perturbed parameter value. We used $r = 1.25$, so that the scale factors ranged over $[1/r, r] = [0.8, 1.25]$. This approach ensured that parameters remained strictly positive and that their increases and decreases were treated as symmetric percentage changes rather than absolute shifts.

For each analysis, we generated a Sobol quasi-Monte Carlo training sample of $2^{13}$ perturbations, ran the simulation, and computed the corresponding EMD values. In addition, we generated and evaluated an independent Sobol validation set of $2^9$ samples with a different random seed. This dataset was reserved exclusively for surrogate validation.

We trained an XGBoost gradient-boosted tree model (*Friedman, 2001*; *Chen and Guestrin, 2016*) as a surrogate (*Storlie et al., 2009*) to approximate the mapping of the perturbed parameters to the EMD to the baseline model. Surrogate accuracy was then assessed on the independent validation set: validation $R^2$ was within 5% of that of training $R^2$ and Spearman rank correlation between simulated and surrogate predictions exceeded 0.95.

Sobol sensitivity indices were then estimated using the surrogate model (*Sobol', 2001*; *Storlie et al., 2009*). We used Saltelli sampling (*Saltelli et al., 2010*), a variance-based Monte Carlo method that generates structured resampling matrices to efficiently compute both first-order and total-order indices. Saltelli sampling was performed with $N*(k+2)$ samples, with $k$ being the number of parameters and $N=10^{14}$. Using the surrogate, we estimated first-order and total-order Sobol indices via quasi-Monte Carlo sampling. Indices are reported with 95% confidence intervals. All indices should be interpreted as variance contributions within the local neighborhood based on the perturbation band [80%,125%], not as global sensitivities across the entire parameter space. Note that reported total-order indices include all interaction terms involving the feature and thus their sum can be greater than 1.

## Acknowledgements

This work was supported by the National Institutes of Health (NIH) grants R01 NS112304, R01 NS115900, R01 NS130799, and T32 NS121768, and the Kentucky Spinal Cord and Head Injury Research Trust.

## Additional information

### Funding

| Funder | Grant reference number | Author |
| --- | --- | --- |
| National Institutes of Health | R01 NS112304 | David SK Magnuson Simon M Danner |
| National Institutes of Health | R01 NS115900 | Simon M Danner |
| National Institutes of Health | R01 NS130799 | Ilya A Rybak |
| National Institutes of Health | T32 NS121768 | Andrew B Lockhart |
| Kentucky Spinal Cord Injury Research Trust | | David SK Magnuson |

The funders had no role in study design, data collection and interpretation, or the decision to submit the work for publication.

### Author contributions

Natalia A Shevtsova, Conceptualization, Formal analysis, Validation, Investigation, Visualization, Methodology, Writing – original draft, Writing – review and editing; Andrew B Lockhart, Validation, Visualization, Writing – review and editing; Ilya A Rybak, Conceptualization, Writing – review and editing; David SK Magnuson, Conceptualization, Funding acquisition, Writing – review and editing; Simon M Danner, Conceptualization, Resources, Data curation, Software, Formal analysis, Supervision, Funding acquisition, Validation, Investigation, Visualization, Methodology, Writing – original draft, Project administration, Writing – review and editing

### Author ORCIDs

Natalia A Shevtsova ⓘ https://orcid.org/0000-0002-1971-9707
Andrew B Lockhart ⓘ https://orcid.org/0000-0002-7141-1347
Ilya A Rybak ⓘ https://orcid.org/0000-0003-3461-349X
David SK Magnuson ⓘ https://orcid.org/0000-0003-3816-3676
Simon M Danner ⓘ https://orcid.org/0000-0002-4642-7064

Reviewer #1 (Public review): https://doi.org/10.7554/eLife.107480.3.sa1
Reviewer #2 (Public review): https://doi.org/10.7554/eLife.107480.3.sa2
Reviewer #3 (Public review): https://doi.org/10.7554/eLife.107480.3.sa3
Author response https://doi.org/10.7554/eLife.107480.3.sa4

## Additional files

### Supplementary files

MDAR checklist

### Data availability

Model configuration files, Python code for running simulations, analyzing results, and processing experimental data, as well as scripts to reproduce all simulations and generate all figures presented in the paper, are available at https://github.com/dannerlab/rat-sci-locomotion-model (copy archived at *Danner et al., 2025*). The neural network simulation package used in this study is available at https://github.com/SimonDanner/CPGNetworkSimulator (copy archived at *Danner, 2025*). The experimental data reproduced from *Danner et al., 2023* can be found at https://github.com/dannerlab/rat-sci-locomotion.

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
