## [Editor Report · eLife Assessment]

This **important** study describes a computational model of the rat spinal locomotor circuits and how they could be plastically reconfigured after lateral hemisection or contusion injuries to replicate gaits observed experimentally *in vivo*. Overall, the simulation results **convincingly** mirror the gait parameters observed experimentally. The model suggests the emergence of detour circuits after lateral hemisection, whereas after a midline contusion, the model suggests plasticity of left-right and sensory inputs below the injury.

---

## [Referee Report · Reviewer #1 (Public review)]

Summary:

This is a rigorous data-driven modeling study extending the authors' previous model of spinal locomotor central pattern generator (CPG) circuits developed for the mouse spinal cord and adapted here to the rat to explore potential circuit-level changes underlying altered speed-dependent gaits due to asymmetric (lateral) thoracic spinal hemisection and symmetric midline contusion. The model reproduces key features of the rat speed-dependent gait-related experimental data before injury and after recovery from these two different thoracic spinal cord injuries and suggests injury-specific mechanisms of circuit reorganization underlying functional recovery. There is much interest in the mechanisms of locomotor behavior recovery after spinal cord injury, and data-driven behaviorally relevant circuit modeling is an important approach. This study represents an important advance of the authors' previous experimental and modeling work on locomotor circuitry and in the motor control field.

Strengths:

(1) The authors use an advanced computational model of spinal locomotor circuitry to investigate potential reorganization of neural connectivity underlying locomotor control following recovery from symmetrical midline thoracic contusion and asymmetrical (lateral) hemisection injuries, based on an extensive dataset for the rat model of spinal cord injury.

(2) The rat dataset used is from an in vivo experimental paradigm involving challenging animals to perform overground locomotion across the full range of speeds before and after the two distinct spinal cord injury models, enabling the authors to more completely reveal injury-specific deficits in speed-dependent interlimb coordination and locomotor gaits.

(3) The model reproduces the rat gait-related experimental data before injury and after recovery from these two different thoracic spinal cord injuries, which exhibit roughly comparable functional recovery, and suggests injury-specific, compensatory mechanisms of circuit reorganization underlying recovery.

(4) The model simulations suggest that recovery after lateral hemisection mechanistically involves partial functional restoration of descending drive and long propriospinal pathways, whereas recovery following midline contusion relies on reorganization of sublesional lumbar circuitry combined with altered descending control of cervical networks.

(5) These observations suggest that symmetrical (contusion) and asymmetrical (lateral hemisection) injuries induce distinct types of plasticity in different spinal cord regions, suggesting that injury symmetry partly dictates the location and type of neural plasticity supporting recovery.

(6) The authors suggest therapeutic strategies may be more effective by targeting specific circuits according to injury symmetry.

Weaknesses:

(1) The recovery mechanisms implemented in the model involve circuit connectivity/connection weights adjustment based on assumptions about the structures involved and compensatory responses to the injury. As the authors acknowledge, other factors affecting locomotor patterns and compensation, such as somatosensory afferent feedback, neurochemical modulator influences, and limb/body biomechanics, are not considered in the model. The authors have now more adequately discussed the limitations of the modeling and associated implications for functional interpretation.

Comments on revisions:

The authors have substantially improved the manuscript by including model parameter sensitivity analyses and by more adequately discussing the limitations of the modeling and associated implications for functional interpretation.

---

## [Referee Report · Reviewer #2 (Public review)]

Summary:

In this paper, the authors present a detailed computational model and experimental data concerning over-ground locomotion in rats before and after recovery from spinal cord injury. They are able to manually tune the parameters of this physiologically based, detailed model to reproduce many aspects of the observed animals' locomotion in the naive case and in two distinct injury cases.

Strengths:

The strengths are that the model is driven to closely match clean experimental data, and the model itself has detailed correspondence to proposed anatomical reality. As such this makes the model more readily applicable to future experimental work. It can make useful suggestions. The model reproduces are large number of conditions, across frequencies, and with model structure changed by injury and recovery. The model is extensive and is driven by known structures, has links to genetic identities, and has been validated extensively across a number of experiments and manipulations over the years. It models a system of critical importance to the field, and the tight coupling to experimental data is a real strength.

Weaknesses:

A downside is that scientifically, here, the only question tackled is one of sufficiency. With manual tuning of parameters in a way that matches what the field believes/knows from experimental work, the detailed model can reproduce the experimental findings. One of the benefits of computational models is that the counter-factual can be tested to provide evidence against alternate hypotheses. That isn't really done here. I'm pretty sure there are competing theories of what happens during recovery from a hemi-section injury and contusion injury. The model could be used to make predictions for some alternate hypothesis, supporting or rejecting theories of recovery. This may be part of future plans. Here, the focus is on showing that the model is capable of reproducing the experimental results at all, for any set of parameters, however tuned.

Comments on revisions:

The authors have addressed my prior concerns and clearly discuss the sufficiency of the model, and strengthen the discussion with interesting findings for the role of propriospinal and commissural interneuronal pathways. This is a very nice contribution.

---

## [Referee Report · Reviewer #3 (Public review)]

Summary:

This study describes a computational model of the rat spinal locomotor circuit and how it could be reconfigured after lateral hemisection or contusion injuries to replicate gaits observed experimentally.

The model suggests the emergence of detour circuits after lateral hemisection whereas after a midline contusion, the model suggests plasticity of left-right and sensory inputs below the injury.

Strengths:

The model accurately models many known connections within and between forelimb and hindlimb spinal locomotor circuits.

The simulation results mirror closely gait parameters observed experimentally. Many gait parameters were studied as well as variability in these parameters in intact versus injured conditions.

A sensitivity analysis provides some sense of the relative importance of the various modified connectivity after injury in setting the changes in gait seen after the two types of injuries

Overall, the authors achieved their aims and the model provides solid support for the changes in connectivity after the two types of injuries modelled. This work emphasizes specific changes in connectivity after lateral hemisection or after contusion that could be investigated experimentally. The model is available to be used by the public and could be a tool used to investigate the relative importance of various highlighted or undiscovered changes in connectivity that could underlie the recovery of locomotor function in spinalized rats.

Comments on revisions:

The authors addressed the comments made by the reviewers. The sensitivity analysis adds insights to the manuscript

---

## [Author Response]

The following is the authors’ response to the original reviews.

**Reviewer #1 (Public review)**:Summary:This is a rigorous data-driven modeling study, extending the authors' previous model of spinal locomotor central pattern generator (CPG) circuits developed for the mouse spinal cord and adapted here to the rat to explore potential circuit-level changes underlying altered speeddependent gaits, due to asymmetric (lateral) thoracic spinal hemisection and symmetric midline contusion. The model reproduces key features of the rat speed-dependent gait-related experimental data before injury and after recovery from these two different thoracic spinal cord injuries and suggests injury-specific mechanisms of circuit reorganization underlying functional recovery. There is much interest in the mechanisms of locomotor behavior recovery after spinal cord injury, and data-driven behaviorally relevant circuit modeling is an important approach. This study represents an important advance in the authors' previous experimental and modeling work on locomotor circuitry and in the motor control field.Strengths:(1) The authors use an advanced computational model of spinal locomotor circuitry to investigate potential reorganization of neural connectivity underlying locomotor control following recovery from symmetrical midline thoracic contusion and asymmetrical (lateral) hemisection injuries, based on an extensive dataset for the rat model of spinal cord injury.(2) The rat dataset used is from an in vivo experimental paradigm involving challenging animals to perform overground locomotion across the full range of speeds before and after the two distinct spinal cord injury models, enabling the authors to more completely reveal injury-specific deficits in speed-dependent interlimb coordination and locomotor gaits.(3) The model reproduces the rat gait-related experimental data before injury and after recovery from these two different thoracic spinal cord injuries, which exhibit roughly comparable functional recovery, and suggests injury-specific, compensatory mechanisms of circuit reorganization underlying recovery.(4) The model simulations suggest that recovery after lateral hemisection mechanistically involves partial functional restoration of descending drive and long propriospinal pathways. In contrast, recovery following midline contusion relies on reorganization of sublesional lumbar circuitry combined with altered descending control of cervical networks.(5) These observations suggest that symmetrical (contusion) and asymmetrical (lateral hemisection) injuries induce distinct types of plasticity in different spinal cord regions, suggesting that injury symmetry partly dictates the location and type of neural plasticity supporting recovery.(6) The authors suggest that therapeutic strategies may be more effective by targeting specific circuits according to injury symmetry.Weaknesses:The recovery mechanisms implemented in the model involve circuit connectivity/connection weights adjustment based on assumptions about the structures involved and compensatory responses to the injury. As the authors acknowledge, other factors affecting locomotor patterns and compensation, such as somatosensory afferent feedback, neurochemical modulator influences, and limb/body biomechanics, are not considered in the model.

We appreciate the positive review and critical comments. We added a dedicate limitation and future direction section (see response recommendations below). Further, we also performed a sensitivity analysis: while the model still relies on a set of hypothesized connectivity changes, this analysis quantifies how robust our conclusions are to these parameter choices and indicates which pathways most strongly affect the recovered locomotor pattern.

**Reviewer #1 (Recommendations for the authors):**
The authors have used an advanced model of rodent spinal locomotor CPG circuits, adapted to the rat spinal cord, which remarkably reproduces the key features of the rat speed-dependent gait-related experimental data before injury and after recovery from the two different thoracic spinal cord injuries studied. Importantly, they have exploited the extensive dataset for the in vivo rat spinal cord injury model involving overground locomotion across the full range of speeds before and after the two distinct spinal cord injuries, enabling the authors to more completely reveal injury-specific deficits in speed-dependent interlimb coordination and locomotor gaits. The paper is well-written and well-illustrated.(1) My only general suggestion is that the authors include a section that succinctly summarizes the limitations of the modeling and points to elaborations of the model and experimental data required for future studies. Some important caveats are dispersed throughout the Discussion, but a more consolidated section would be useful.

We added a dedicated Limitations and future directions section that consolidates shortcomings and broadly outlines potential next steps in terms of modeling and experimental data. Specifically, we highlight the issue of lack of afferent feedback connections in the model, lack of consideration of biomechanic mechanisms, and restriction of the model to beneficial plasticity. To resolve these issues, we need neuromechancial models (integration of the neural circuits with a model of the musculoskeletal system), experimental data validating our predictions and data to constrain future models to be able to distinguish between beneficial and maladaptive plasticity.

(2) Please correct the Figure 11 legend title to indicate recovery after contusion (not hemisection).

Done. Thanks for noticing.

**Reviewer #2 (Public review):**
Summary:In this paper, the authors present a detailed computational model and experimental data concerning overground locomotion in rats before and after recovery from spinal cord injury. They are able to manually tune the parameters of this physiologically based, detailed model to reproduce many aspects of the observed animals locomotion in the naive case and in two distinct injury cases.Strengths:The strengths are that the model is driven to closely match clean experimental data, and the model itself has detailed correspondence to proposed anatomical reality. As such, this makes the model more readily applicable to future experimental work. It can make useful suggestions. The model reproduces a large number of conditions across frequencies, and with the model structure changed by injury and recovery. The model is extensive and is driven by known structures, with links to genetic identities, and has been extensively validated across multiple experiments and manipulations over the years. It models a system of critical importance to the field, and the tight coupling to experimental data is a real strength.Weaknesses:A downside is that, scientifically, here, the only question tackled is one of sufficiency. By manually tuning parameters in a manner that aligns with the field's understanding from experimental work, the detailed model can accurately reproduce the experimental findings. One of the benefits of computational models is that the counterfactual can be tested to provide evidence against alternative hypotheses. That isn't really done here. I'm fairly certain that there are competing theories regarding what happens during recovery from a hemi-section injury and a contusion injury. The model could be used to make predictions for some alternative hypotheses, supporting or rejecting theories of recovery. This may be part of future plans. Here, the focus is on showing that the model is capable of reproducing the experimental results at all, for any set of parameters, however tuned.

We agree with the reviewer that the present study focuses on sufficiency, and we now explicitly acknowledge this in the revised limitations section. We also added sensitivity analysis (for details see response to reviewer 3) that provides an initial assessment of robustness to the assumed connectivity changes. We note that the model reproduces a broad set of experimentally observed features across the full range of locomotor frequencies (including loss and emergence of specific gaits, reduced maximum stepping frequency, and altered variability of interlimb phase differences) using only a small set of hypothesized circuit reorganizations that have been experimentally observed but previously only correlated with recovery. Our results therefore suggest that this limited set of changes is indeed sufficient to account for the complex pattern of recovered locomotor behavior.

Finally, although exploring alternative solutions is of interest, we believe such efforts will be most informative once afferent feedback is incorporated, which we see as the logical next step in our studies.

**Reviewer #2 (Recommendations for the authors):**
The paper could be strengthened with some more scientific interpretation and future directions. What are some novel predictions that can be made with the model, now that it has shown sufficiency here, that could guide future experimental work? Does it contradict in any way theories of CPG structure or neuronal plastic recovery?

The sensitivity analysis that we performed in response to reviewer 3’s suggestion expanded our interpretation/conclusions by showing that, although injury symmetry (contusion vs. lateral hemisection) influences *which* pathways reorganize, recovered locomotion across injury type depends most strongly on restored activation of lumbar rhythm-generating and strengths of lumbar commissural circuits.

Interestingly, this sensitivity analysis also showed that variations of strengths of long propriospinal pathways (ascending, descending, spared, injured-and-recovered) have a much smaller, almost negligible effect, when compared to variations of drive to lumbar rhythm generators or lumbar commissural interneuron connection weights in the same range (see Fig 13, 13-supplement 1 and 2). This is in accordance with our initial model suggestions that after contusion LPN connections weight had to be lowered to values substantially lower than what was expected by the severity of the injury. Which is also corroborated by our anatomical findings that in parallel to recovery from contusion, the number of synaptic connections by LAPNs to the cervical enlargement were reduced, and that silencing of LPNs post-contusion improves locomotion. These surprising findings have been extensively discussed in the discussion section.

Together, these findings suggest that experimental characterization of reorganization of the lumbar circuitry with a specific focus on commissural interneurons and inputs to the lumbar circuitry that could restore activation of sublesional lumbar rhythm generators is a crucial next step for understanding post-injury plasticity and recovery of locomotor function. This is now clearly discussed.

Finally, we note that a key contribution of this work is that the model demonstrates a plausible mechanistic link between specific circuit reorganizations and recovered locomotor function, a relationship previously supported mainly by correlative evidence.

**Reviewer #3 (Public review):**
Summary:This study describes a computational model of the rat spinal locomotor circuit and how it could be reconfigured after lateral hemisection or contusion injuries to replicate gaits observed experimentally.The model suggests the emergence of detour circuits after lateral hemisection, whereas after a midline contusion, the model suggests plasticity of left-right and sensory inputs below the injury.Strengths:The model accurately models many known connections within and between forelimb and hindlimb spinal locomotor circuits.The simulation results mirror closely gait parameters observed experimentally. Many gait parameters were studied, as well as variability in these parameters in intact versus injured conditions.Weaknesses:The study could provide some sense of the relative importance of the various modified connectivities after injury in setting the changes in gait seen after the two types of injuries.

We performed a local sensitivity analysis of the hemisection and contusion models to identify which connectivity changes most strongly influence post-injury locomotor behavior. Key parameters (descending drive to sublesional rhythm generators and the strength of selected commissural and propriospinal pathways) were perturbed within 80–125% of their baseline values, and for each perturbation we quantified changes in model output using the Earth Mover’s Distance between baseline and perturbed simulations in a 7-dimensional space (six interlimb phase differences plus locomotor frequency). We then trained a surrogate model and computed Sobol first- and total-order sensitivity indices, which quantify how much each parameter and its interactions contribute to variability in this distance measure. This analysis showed that, across both injuries, variations in drive to sublesional lumbar rhythm generators and in lumbar V0/V3 commissural connectivity have the largest impact on recovered gait expression, whereas other pathways had comparatively minor effects within the tested range.

The sensitivity analysis further refined our conclusions by showing that, although injury symmetry (contusion vs. lateral hemisection) influences *which* pathways reorganize, effective recovery in both cases depends on re-engaging lumbar rhythm-generating and commissural circuits, highlighting these networks as key therapeutic targets.

Overall, the authors achieved their aims, and the model provides solid support for the changes in connectivity after the two types of injuries were modelled. This work emphasizes specific changes in connectivity after lateral hemisection or after contusion that could be investigated experimentally. The model is available for public use and could serve as a tool to analyze the relative importance of various highlighted or previously undiscovered changes in connectivity that may underlie the recovery of locomotor function in spinalized rats.
**Reviewer #3 (Recommendations for the authors):**
(1) It would be useful to study the sensitivity of the injured models to small changes in the connectivity changes to determine which ones play a greater role in the gait after injury.

See response above on the added sensitivity analysis.

(2) Was there any tissue analysis from the original experiments with the contusion experiments, as contusion experiments can be variable, so it would be good to know the level of variability in the injuries?

Unfortunately, we were unable to complete tissue analysis of the injury epicenters for these animals because the tissue was not handled appropriately for histology. However, in the past, comparable animals with T10 12.5g-cm contusion injuries delivered by the NYU (MASCIS) Impactor had variability of up to ~30% of the mean (spared white matter, e.g. see Smith et al., 2006). It is also worth noting that spared white matter at the epicenter, at least in our hands, is generally well-correlated with BBB overground locomotor scale scores.

(3) There is more variability in phase difference in rats than model in the lateral hemisection. Is there any way to figure out which of the connectivity changes is most responsible for that variability?

We agree that the variability of phase differences after lateral hemisection is larger in rats than in the model. One possible contributor to this discrepancy is the strength of spared long propriospinal neuron (LPN) pathways, which we kept fixed at pre-injury levels in the model. As an exploratory analysis, we varied the weights of these spared LPN connections and quantified the circular standard deviation of the phase differences (Author response image 1). Decreasing spared LPN weights increased the variability of all phase differences. This suggests that plasticity of spared LPNs (potentially reducing their effective connectivity and partly compensating for the asymmetry introduced by the lesion) could contribute to the higher variability seen in vivo. However, because these results remain speculative, we chose to include them in this response only and not in the main manuscript.

**Author response image 1. sa4fig1:** Variability of phase differences as a function of spared long propriospinal neuron connection weights (hemisection model).